# Seasonal ecosystem vulnerability to climatic anomalies in the Mediterranean

Johannes Vogel[1,2], Eva Paton[2], and Valentin Aich[3]

[1]Institute of Environmental Science and Geography, University of Potsdam, Potsdam, Germany
[2]Institute of Ecology, Technical University of Berlin, Berlin, Germany
[3]Global Water Partnership, Geneva, Switzerland

**Correspondence:** Johannes Vogel (joschavogel@uni-potsdam.de)

**Abstract.** Mediterranean ecosystems are particularly vulnerable to climate change and the associated increase in climate anomalies. This study investigates extreme ecosystem responses evoked by climatic drivers in the Mediterranean Basin for the time span 1999–2019 with a specific focus on seasonal variations, as the seasonal timing of climatic anomalies is considered essential for impact and vulnerability assessment. A bivariate vulnerability analysis is performed for each month of the year to quantify which combinations of the drivers temperature (obtained from ER5 Land) and soil moisture (obtained from ESA CCI and ERA5 Land) lead to extreme reductions of ecosystem productivity using the fraction of absorbed photosynthetically active radiation (FAPAR; obtained from Copernicus Global Land Service) as a proxy.

The bivariate analysis clearly showed that, in many cases, it is not just one but a combination of both drivers that causes ecosystem vulnerability. The overall pattern shows that Mediterranean ecosystems are prone to three soil moisture regimes during the yearly cycle: They are vulnerable to hot and dry conditions from May to July, to cold and dry conditions from August to October, and to cold conditions from November to April, illustrating the shift from a soil moisture-limited regime in summer to an energy-limited regime in winter. In late spring, a month with significant vulnerability to hot conditions only often precedes the next stage of vulnerability to both hot and dry conditions, suggesting that high temperatures lead to critically low soil moisture levels with a certain time lag. In the eastern Mediterranean, the period of vulnerability to hot and dry conditions within the year is much longer than in the western Mediterranean. Our results show that it is crucial to account for both spatial and temporal variability to adequately assess ecosystem vulnerability. The seasonal vulnerability approach presented in this study helps to provide detailed insights regarding the specific phenological stage of the year in which ecosystem vulnerability to a certain climatic condition occurs.

## 1 Introduction

Drought frequency and intensity is increasing in the Mediterranean, accompanied by rising temperatures and heat wave intensities (Perkins-Kirkpatrick and Gibson, 2017; Samaniego et al., 2018; IPCC, 2019; Tramblay et al., 2020). These climatic changes are linked to vulnerability of ecosystems in various ways, e.g. to reductions in forest growth and increasing tree mortality (Sarris et al., 2007, 2011), as well as extended fire risk (Sarris et al., 2014; Ruffault et al., 2018) and declining agricultural yields (Peña-Gallardo et al., 2019; Fraga et al., 2020). Furthermore, the ability to provide ecosystem services is

impaired due to alterations in functioning and structure of Mediterranean ecosystems,(Ogaya and Peñuelas, 2007; Peñuelas et al., 2017). Broad-scale vegetation shifts and replacement of species are projected and ultimately desertification is expected in many Mediterranean regions (Gao and Giorgi, 2008; Zdruli, 2011; Feng and Fu, 2013; Liu et al., 2018).

The Mediterranean climate is characterised by great spatial and temporal variability, which makes the investigation of ecosystem impacts challenging. The Mediterranean Basin is marked by complex topography and is influenced by several large-scale

atmospheric patterns (Lionello et al., 2006, 2012). Furthermore, the Mediterranean climate has an intricate seasonal cycle, alternating between water-limited conditions in summer and energy-limited conditions in winter (Spano et al., 2013). An assessment of ecosystem vulnerability in the Mediterranean therefore needs to account for both its spatial and temporal variability.

In this study, we build on the ecosystem vulnerability analysis proposed by van Oijen et al. (2013, 2014); Rolinski et al. (2015), adapted with a focus on seasonal and multivariate impacts using remote sensing and reanalysis data. We enhance the

ecosystem vulnerability concept with a focus on the seasonal timing of impacts. Ecosystem responses differ depending on the seasonal timing of the event (de Boeck et al., 2011; Smith, 2011; Sippel et al., 2016). Shifts of only a few weeks in drought occurrence can make the difference between negligible and detrimental impacts (Denton et al., 2017; Sippel et al., 2017, 2018). Even though accounting for seasonality is crucial in investigating climatic impacts on ecosystems, it is still often neglected (Piao et al., 2019). Studies are frequently limited to particular periods of interest within the year – usually a period of up to

half year centered around summer – when investigating seasonality (van Oijen et al., 2014; Baumbach et al., 2017; Nicolai-Shaw et al., 2017; Karnieli et al., 2019), but rarely investigate the seasonality year-round. In addition, combinations of climatic events in the seasonal cycle are seldom addressed (Smith, 2011; Hatfield and Prueger, 2015). Due to the pronounced land-atmosphere feedback mechanisms in the Mediterranean (Seneviratne et al., 2006; Green et al., 2017; Tramblay et al., 2020), it is particularly important to analyse the impacts of climatic anomalies in soil moisture and temperature jointly rather than

in isolation (Mueller and Seneviratne, 2012). Such joint impacts of multiple stressors on ecosystems are still little researched (Shukla et al., 2019). Relationships between climatological and ecological variables at the tails of the distribution can show distinctly different behaviour compared to the findings based on conventional linear correlation, which makes it especially important to investigate the impact of climate anomalies on ecosystems, not only their mean behaviour (Jentsch et al., 2007; Reyer et al., 2013; Baumbach et al., 2017; Ribeiro et al., 2020).

Soil moisture is a particularly relevant variable for assessing the state of ecosystems as it is directly related to plant activity, biomass and agricultural yields (McWilliam, 1986; Sherry et al., 2008; Seneviratne et al., 2010; Zscheischler et al., 2013), especially in seasonally water-limited areas such as the Mediterranean (Szczypta et al., 2014). However, large-scale soil moisture data covering long time spans is scarce. Therefore, soil moisture proxies are applied in most cases, e.g. land surface models or drought indicators such as the SPI (Dorigo et al., 2017; Nicolai-Shaw et al., 2017). However, the SPI is primarily an in-

dicator for meteorological droughts, which do not necessarily propagate into soil moisture droughts (de Boeck et al., 2011). Only a few studies use soil moisture data derived from satellite imagery because long-term coverage was not available until recently. Individual satellites do not cover sufficiently long time spans, but long-term coverage can be achieved by merging soil moisture data from several satellites. The European Space Agency's Climate Change Initiative (ESA CCI) soil moisture data set provides a unique, globally consistent multi-decadal time series based on several active and passive microwave sensors

(Dorigo and de Jeu, 2016; Dorigo et al., 2017). It was first published in 2012 and has continuously improved since (Dorigo et al., 2017). It has proven capability to assess land-vegetation-atmosphere dynamics (de Jeu and Dorigo, 2016; Dorigo and de Jeu, 2016; Nicolai-Shaw et al., 2017; Gruber et al., 2019). So far, satellite-based soil moisture data are still rarely used in ecosystem research (Dorigo et al., 2017), and e.g. Rolinski et al. (2015) point out the need to use observational data in the assessment of ecosystem vulnerability. Therefore, we seek to put greater emphasis on the possibilities arising from newly

available remote-sensing products within the last years. In addition, we also performed the analysis using the soil moisture product from the ERA5 Land reanalysis data set. The fraction of absorbed photosynthetically active radiation (FAPAR) is used as an indicator of ecosystem productivity in our study. The FAPAR is crucial for monitoring climatic impacts on terrestrial ecosystems and is directly related to the photosynthetic activity of vegetation and thus to its greenness and health (Potter et al., 2003; Gobron et al., 2010; Ivits et al., 2016). Vegetation indices such as the Normalized Difference Vegetation Index (NDVI)

are closely related to the FAPAR and can be seen as proxies (Myneni and Williams, 1994; Pinty et al., 2009).

This study aims to quantify ecosystem vulnerability by assessing which combinations of climatic drivers lead to extreme reductions in ecosystem productivity in the Mediterranean Basin using a bivariate vulnerability analysis with a specific focus on seasonal variations. Soil moisture and temperature are investigated as climatic drivers, and the FAPAR is used to assess the ecological response. Furthermore, ecosystem vulnerability is calculated separately by land cover class and subregion to

account for the spatial complexity of the Mediterranean Basin.

## 2 Methods

### 2.1 Study area

The study area is constrained to all grid points in the Mediterranean Basin belonging to the Köppen-Geiger classes Csa ("Warm temperate climate with dry and hot summer") and Csb ("Warm temperate climate with dry and warm summer")(cf. Fig. 1) to

ensure a certain level of comparability within the study area. Furthermore, the study area is subdivided into land cover classes and subregions. The land cover classes were aggregated according to Table B1 using the ESA CCI land cover classification map of 2018. Grid points where the land cover changed between 1999 and 2018 were excluded in this study, as well as grid points belonging to the land cover classes "Water bodies" and "Urban areas". The countries belonging to each subregion are listed in Table B2.

### 2.2 Data

Daily satellite-based soil moisture data from ESA CCI was obtained at a resolution of 0.25° from 1978–2019 (Gruber et al., 2019). The merged data set (v04.7), containing data from both active and passive sensors, is used. The quality of this data set has continuously improved over the years due to the incorporation of an increasing number of satellites (Dorigo et al., 2017). The data set is representative for the topsoil surface layer of up to 2 cm thickness (Kidd and Haas, 2018). Monthly air

temperature and soil moisture reanalysis data are retrieved from ERA5 Land produced by the European Centre for Medium-

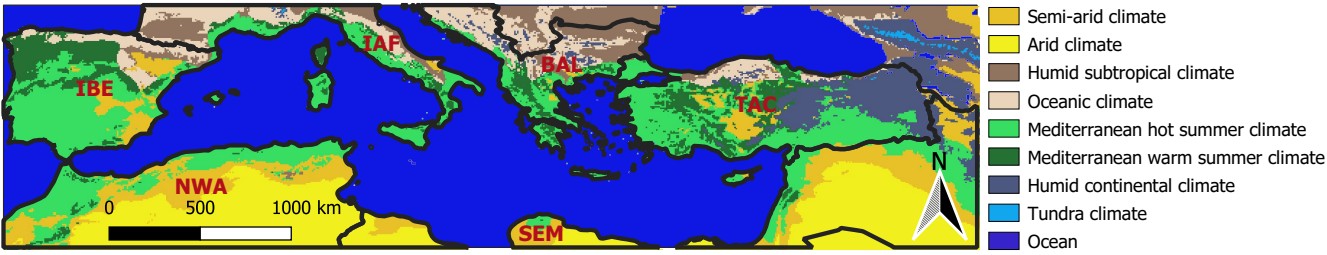

**Figure 1.** Study area in the Mediterranean Basin: The Köppen-Geiger climate categories "Mediterranean hot summer climate" (light green) and "Mediterranean warm summer climate" (dark green) are included in this study. The study area was divided into six subregions: the Iberian Peninsula (IBE), Italy and France (IAF), the Balkan Peninsula (BAL), Turkey and Cyprus (TAC), the southeastern Mediterranean (SEM) and northwestern Africa (NWA)

Range Weather Forecasts (ECMWF) at a resolution of 9 km from 1981–2019. (Copernicus Climate Change Service, 2019). The three soil moisture layers corresponding to the depths 0–7 cm, 7–28 cm and 28–100 cm are used in the analysis. This study is conducted using the ESA CCI soil moisture data set, as well as the ERA5 Land soil moisture data set to verify the robustness of our results. The FAPAR is obtained from the Copernicus Global Land Service (CGLS) (Baret et al., 2013; Verger
et al., 2014). It is derived from SPOT/VGT from 1999–2013 and PROBA-V from 2014–2019 and is provided in ten-day steps (Verger et al., 2019). Furthermore, the ESA CCI land cover classification for the years 1999 and 2018 with a spatial resolution of 300 m (v2.1.1) was used (ESA, 2017). The Köppen-Geiger classification map was acquired from Kottek et al. (2006) and Rubel et al. (2017).

### 2.3 Data preprocessing

All data sets are resampled to a common spatial and temporal resolution of 0.25° and a monthly time step, respectively. The investigated time span encompasses 21 years from 1999–2019. Grid points with more than 60 months of missing soil moisture data within the period from 1999–2019 were excluded from this study (see Fig. 2). These are primarily grid points located close to the coast. In a next step, all variables are deseasonalised by subtracting the annual cycle to account for extremeness relative to the respective time of the year. The variables are z-transformed by subtracting the monthly mean and dividing by the
year-round standard deviation of the deseasonalised time series (Eq. (1)). Z-score transformation allows for a direct comparison of values despite their different physical units (Orth et al., 2020).

$$z_i = \frac{X_i - \mu_{i,month}}{\sigma_i} \tag{1}$$

The impact of environmental drivers on ecosystems may show a time lag of up to a few months – so called "legacy effects" (von Buttlar et al., 2018; Piao et al., 2019). Hence, a moving average of three months $n = 3$ is applied to the environmental
driver variables $env$ temperature and soil moisture, i.e. the preceding two months are included with equal weight for each

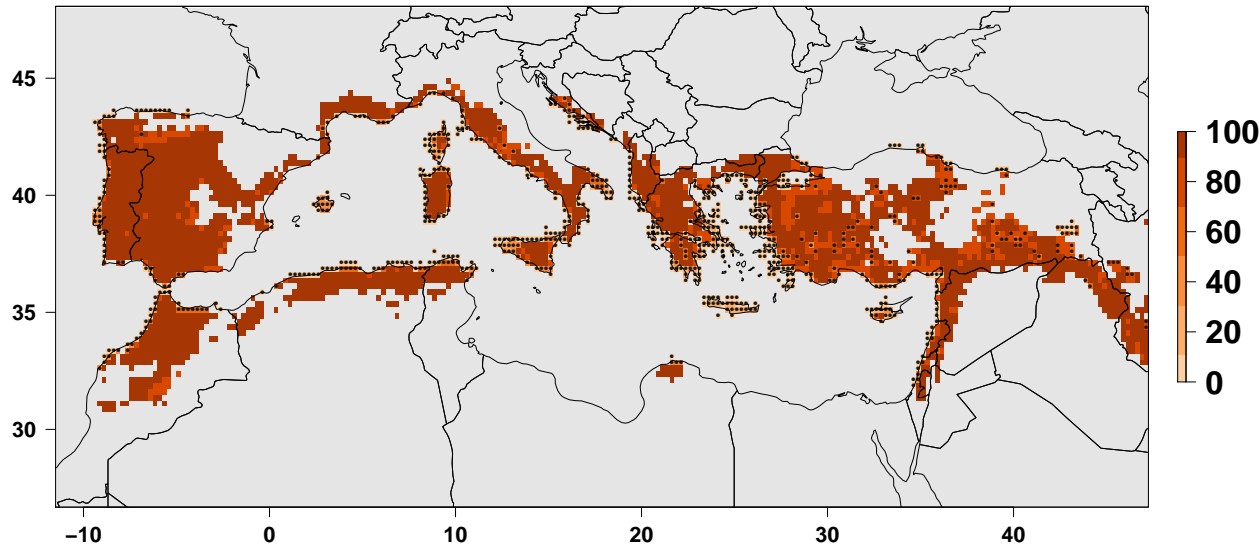

**Figure 2.** Percentage of available monthly soil moisture values from 1999–2019. All grid points excluded from this study are marked with a dot.

monthly time step $i$ in the time span of $m = 21$ years (Eq. (2)) to account for lagged effects.

$$env_i = \frac{1}{n} \sum_{k=i-2}^{i} env_k \text{ for } i \in (1,...,m \times 12) \tag{2}$$

## 2.4 Derivation of ecosystem vulnerability

In the context of our study, ecosystem vulnerability depicts if ecosystems are susceptible or sensitive to a certain hazard. It
allows to attribute states of low ecosystem productivity to certain climatic conditions by linking such states to corresponding deviations in temperature and soil moisture. The terminology on ecosystem vulnerability is confusing since several partially ambiguous terms exist due to the concept being still rather new in ecological research (van Oijen et al., 2013; Weißhuhn et al., 2018). Following the definition by Rolinski et al. (2015), "ecosystem vulnerability $V_E$ is the average deviation of the environmental variable under hazardous ecosystem conditions from values under non-hazardous ecosystem conditions" in our
approach. Here, the environmental variable $env$ is either temperature or soil moisture, respectively, and the ecosystem variable $sys$ is the FAPAR. Ecosystem vulnerability $V_E$ is calculated according to Eq. (3) as the difference of the expectation value

$E_\text{nonhaz}$ of the environmental variable $env$ under non-hazardous conditions of the ecosystem variable $sys$ and the respective value $E_\text{haz}$ under hazardous conditions of the ecosystem variable $sys$ (van Oijen et al., 2013; Rolinski et al., 2015).

$$V_E = E(env|sys\,\text{nonhaz}) - E(env|sys\,\text{haz}) \tag{3}$$

with conditional expectational values defined following Eq. (4)

$$E(env|\circ) = \int env\,\mathbb{P}(env|\circ)\,\mathrm{d}\,env \tag{4}$$

where $\mathbb{P}$ is the probability of $env$ under the specified condition $\circ$ ($sys\,\text{nonhaz}$ or $sys\,\text{haz}$). The probability of hazard occurrence $\mathbb{P}_H$ is given by the number of data points under hazardous conditions $N_\text{haz}$ divided by the total number of data points N, which gives $\mathbb{P}_H = N_\text{haz}/N$. The discrimination threshold between non-hazardous and hazardous ecosystem conditions is set as the 130   10$^\text{th}$ percentile of the FAPAR values for each grid point individually, i.e. $\mathbb{P}(sys\,\text{haz})$ is fixed to 0.1 in this study. Such a threshold is commonly used in ecoclimatological studies (Ahlström et al., 2015; Baumbach et al., 2017; Nicolai-Shaw et al., 2017). To investigate the robustness of our results, we also performed the analysis using the 5$^\text{th}$ and 15$^\text{th}$ percentile for discrimination of hazardous and non-hazardous ecosystem conditions. The spatial and temporal patterns for these cases were in agreement with the 10$^\text{th}$ percentile chosen in our study (results not shown), which indicates that our results are not sensitive to the choice 135   of the percentile. Every grid point has the same number of months with hazardous ecosystem conditions, i.e. the same risk of exceeding the threshold is assumed uniformly for all grid points.

We used the Mann-Whitney U test to investigate significant deviations of climatic conditions during non-hazardous and hazardous ecosystems conditions, which was adjusted for multiple testing using the Benjamini and Hochberg (1995) correction. Significant positive values indicate ecosystem vulnerability $V_E$ to cold (dry) conditions for the climatic driver temperature (soil 140   moisture). Similarly, significant negative values are associated with vulnerability to hot (wet) conditions. In the case of two climatic drivers, this leads to nine possible vulnerability conditions (see Fig. 3). The corresponding p-values are not shown throughout the article due to the large amount of data. A schematic display of the calculation of ecosystem vulnerability $V_E$ is given in Fig. 4 for an exemplary grid point with vulnerability to hot and dry conditions for the month of July. The two drivers temperature and soil moisture are assessed for their effects on ecosystem vulnerability. In this example, the average 145   temperature in July during non-hazardous ecosystem conditions $E_\text{nonhaz}$ is lower than the average during hazardous ecosystem conditions $E_\text{haz}$, leading to a negative vulnerability to temperature, i.e. vulnerability to hot conditions (Fig. 4(a)). For soil moisture, the average soil moisture during non-hazardous ecosystem conditions $E_\text{nonhaz}$ is higher than soil moisture during hazardous conditions $E_\text{haz}$, therefore vulnerability is positive, indicating vulnerability to dry conditions (Fig. 4(b)). Our approach is impact-based, i.e. it focusses on the extremeness of the impact rather than the extremeness of the driver as this enables 150   relating multiple drivers to a single outcome (Zscheischler et al., 2014, 2018). According to the framework by Smith (2011), vulnerability to extreme climatic events is defined as a climate extreme leading to an extreme ecological response. Therefore, our definition differs in that regard that it comprises extremeness only for the ecological response, not necessarily for the climatic driver. The definition used here is broader than the one by Smith (2011), because it includes significant deviations of

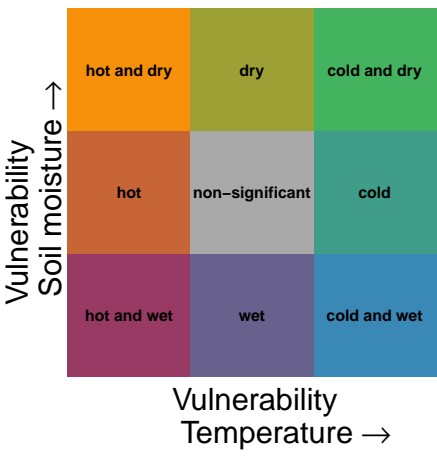

**Figure 3.** Illustration of the vulnerability to all potentially occurring climatic conditions.

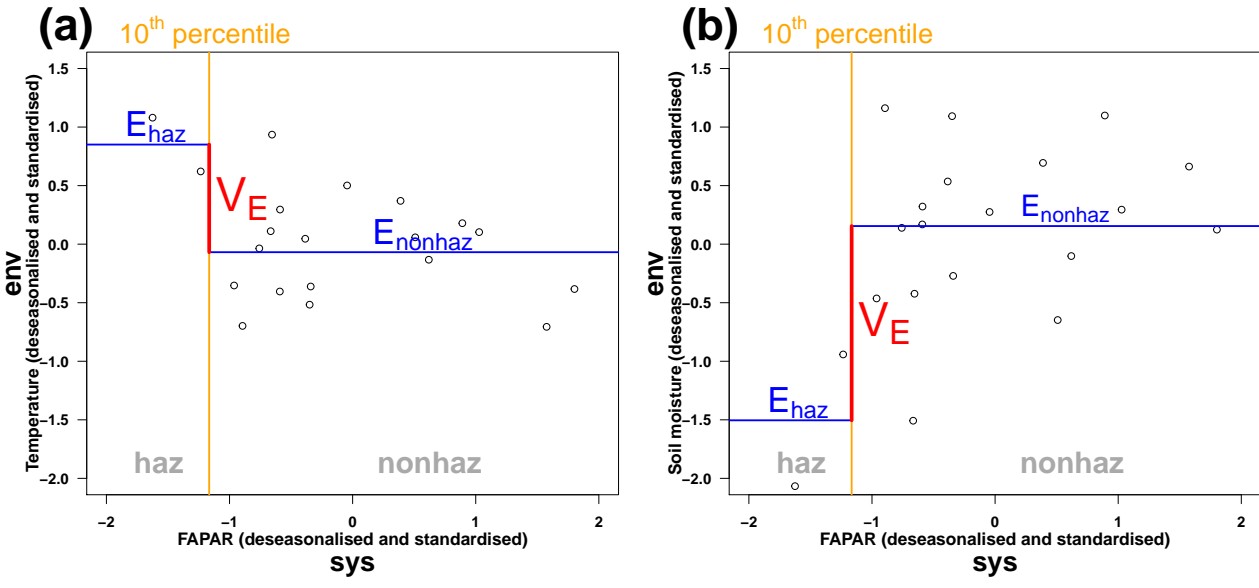

**Figure 4.** Schematic display of ecosystem vulnerability $V_E$ for an exemplary grid point for (a) temperature and (b) soil moisture as environmental driver for the month of July.

the driver variable in general, not only extremes. In our case, ecosystem vulnerability rather shows if the ecosystem variable is
susceptible to certain climatic conditions (which do not need to be extreme). The analysis was carried out using R version 3.6 and Climate Data Operators (CDO) version 1.9 (Schulzweida, 2019; R Core Team, 2020).

## 3 Results

### 3.1 Ecosystem vulnerability by land cover

Figure 5 displays the ecosystem vulnerability to soil moisture and temperature for each land cover class and each month of the year, as well as the corresponding statistical significance indicated by the background colour (see explanation in Fig. 3). The vulnerability to temperature and soil moisture can be summarised into three major regimes during the course of the year (see Fig. 5). From May to July, the vegetation is especially prone to hot and dry conditions. From August to October, there is a shift to a vulnerability to cold and dry conditions in general. Finally, from November to April cold and wet conditions are usually associated with high vulnerability of the vegetation. There are sharp transitions in ecosystem vulnerability from April to May, from July to August, and from October to November for most land cover classes.

In the period from November to March the vast majority of land covers is vulnerable to cold conditions. From March to May there is a transition phase from cold to hot conditions. While in March almost all land covers are vulnerable to cold conditions, in April only four of them still remain vulnerable ("Forest (broadleaved)", "Forest (needleleaved)", " Mixed" and "Shrubland") and none are vulnerable in May, when the majority shifts to vulnerability to hot conditions. In summer, a period with significant vulnerability to hot conditions only precedes the next phase of vulnerability to both hot and dry conditions, e.g. for "Crops (rainfed)" and "Grassland", indicating that the heat desiccates the soil first until it reaches critically low soil moisture levels in the following months. The cycle reverses around July and August. While four land cover classes are still vulnerable to hot conditions in July, none of the classes are in August. Vulnerability to high temperatures is almost entirely restricted to the period from May to July. From August to October, most land cover classes exhibit vulnerability to cold and dry conditions, and from midsummer to the beginning of autumn almost all land cover classes are prone to drought. In the following period from November to March, cold and wet conditions prevail on average. The vulnerability to wet conditions is highest from November to January, whereas many land cover classes are insensitive to soil moisture during most of the time from February to May. Exemptions are e.g. "Forest (broadleaved)", "Crops (rainfed)" and "Mixed", where low ecosystem productivity coincides with wet conditions e.g. in March to April.

The vulnerability to hot conditions of "Grassland" is one month ahead of most other land classes, starting already in April. This could indicate a faster response of this land cover class to environmental drivers than other land cover classes. Sparse vegetation is probably well adapted to high temperatures, as it never shows vulnerability to hot conditions, which means that temperature during extreme ecosystem conditions is not significantly higher than during non-extreme ecosystem conditions. It also never coincides with significantly wet conditions, which might point out that transpiration in these areas is never so high that it could contribute substantially to the desiccation of the soil and thus its influence on soil moisture is negligible.

### 3.2 Ecosystem vulnerability by subregions

Similarly to Fig. 5, ecosystem vulnerability for each subregion is shown in Fig. 6. There is more variability than regarding land cover classes and the general pattern of most land cover classes with a "hot and dry" regime followed by a "cold and dry" regime and subsequently by a "cold and wet" regime does not hold true for most of the Mediterranean subregions. The

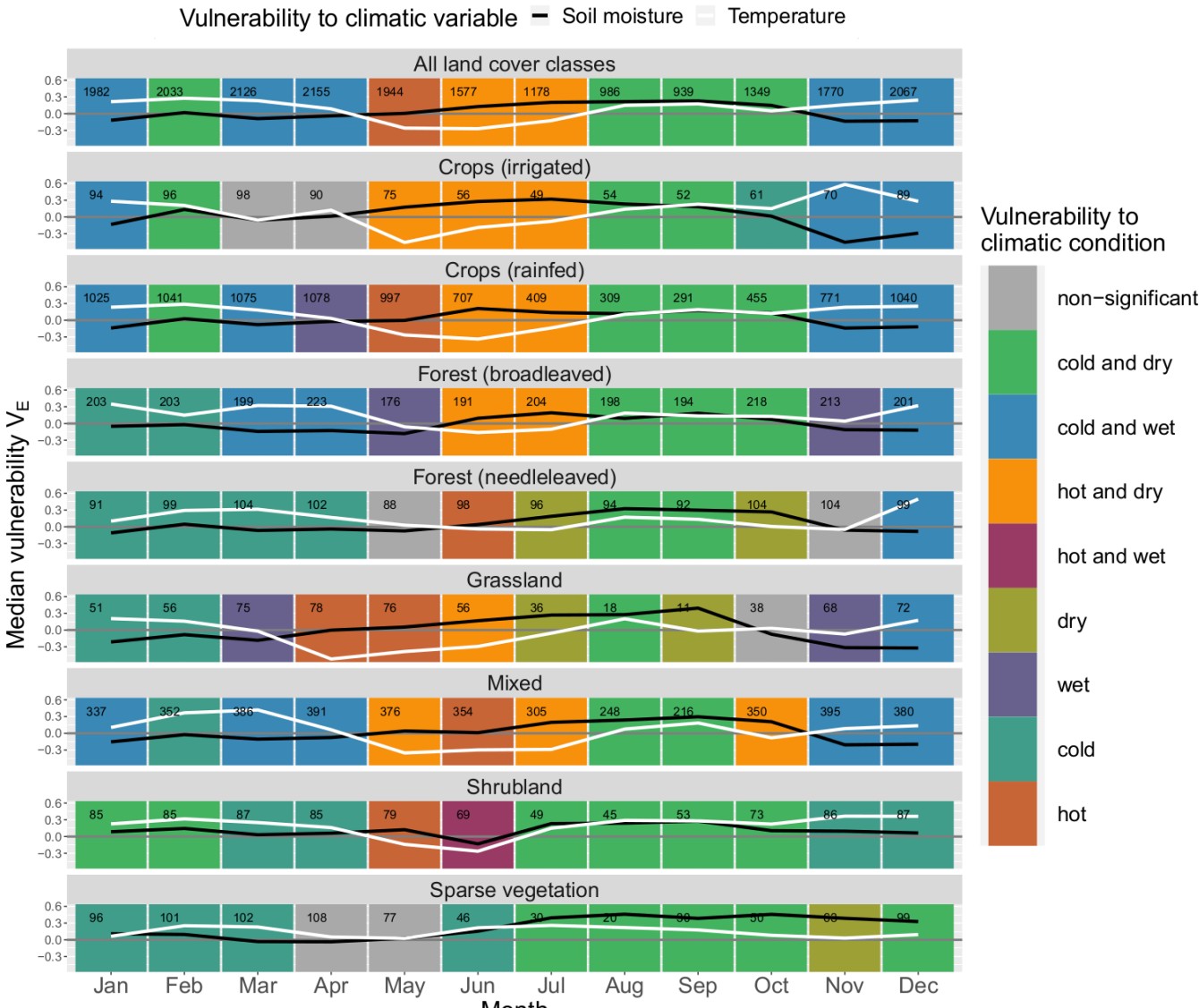

**Figure 5.** Median monthly ecosystem vulnerability per land cover: Vulnerability to temperature (ERA5 Land) is shown in white and vulnerability to soil moisture (ESA CCI) is shown in black for each month of the year (columns) for each land cover (rows). Months with statistically significant deviation of climatic drivers during non-hazardous and hazardous ecosystems conditions according to the Mann-Whitney U test based on a significance level $\alpha = 0.05$ are shown in colour (see legend), all other months are shown in grey. The number of grid points in which an event has occurred in this month and land cover within the period 1999-2019 is shown in the upper left corner of each panel.

vulnerability to soil moisture usually peaks during summer or autumn and reaches a minimum in spring or winter – exceptions

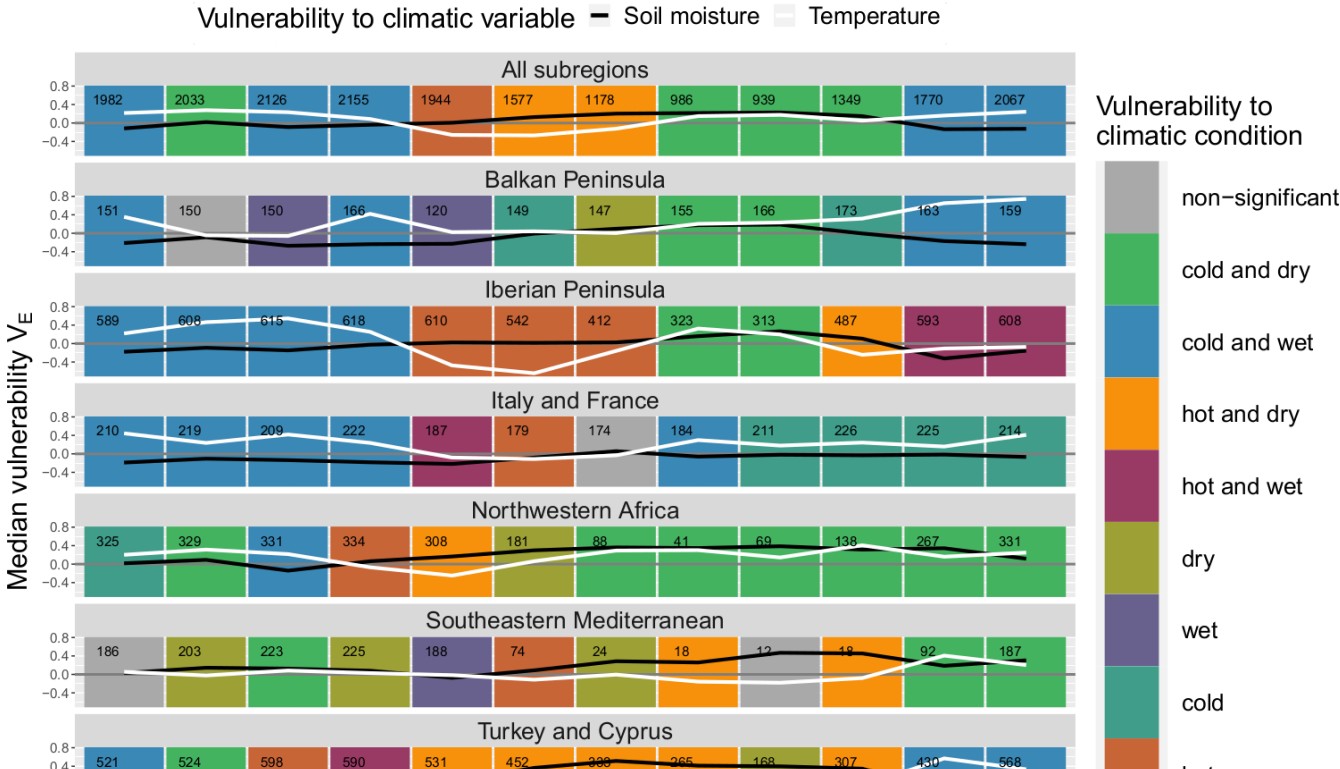

**Figure 6.** Median monthly ecosystem vulnerability per subregion: Vulnerability to temperature (ERA5 Land) is shown in white and vulnerability to soil moisture (ESA CCI) is shown in black for each month of the year (columns) for each land cover (rows). Months with statistically significant deviation of climatic drivers during non-hazardous and hazardous ecosystems conditions according to the Mann-Whitney U test based on a significance level $\alpha = 0.05$ are shown in colour (see legend), all other months are shown in grey. The number of grid points in which an event has occurred in this month and subregion within the period 1999-2019 is shown in the upper left corner of each panel.

are Italy and France as well as the southeastern Mediterranean. The yearly development of vulnerability to temperature is characterised by a minimum around late spring or summer.

There is an extended period of time in which ecosystems are prone to hot conditions from March to October in Turkey, whereas in other regions this period often only lasts for two to three months in spring and summer. Northwestern Africa and the southeastern Mediterranean are prone to dry conditions nine and eight months of the year, respectively, indicating that these regions are usually soil-moisture limited. Italy and France have the lowest sensitivity to soil moisture with only small deviations from zero. Nevertheless, these deviations are significant for half of the months in the year. Interestingly, the Balkan Peninsula

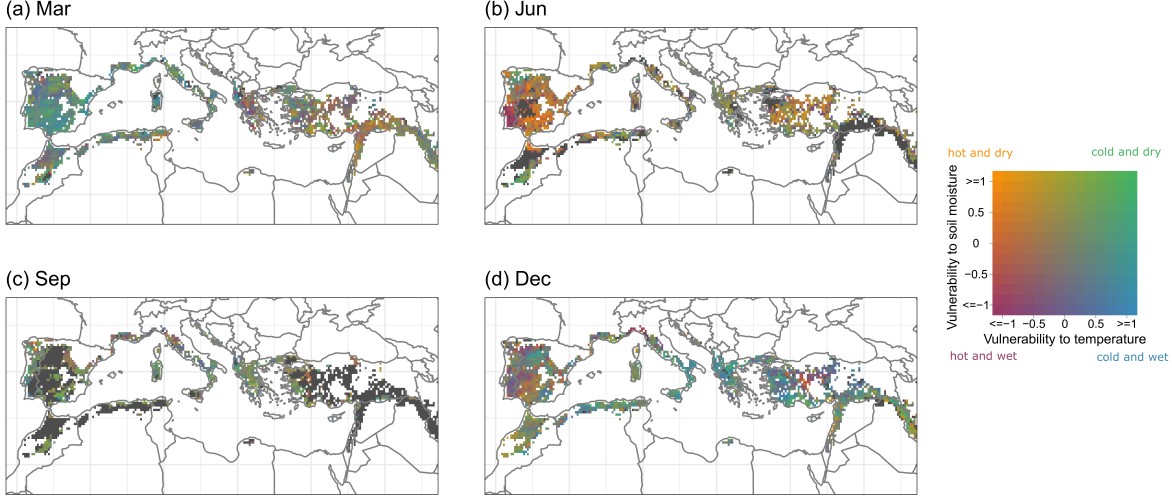

**Figure 7.** Average monthly vulnerability to soil moisture (ESA CCI) and temperature (ERA5 Land) in the Mediterranean Basin for (a) March, (b) June, (c) September and (d) December. grid points without any events during the respective month are displayed black.

is never prone to hot conditions. Outside of the summer season, wet conditions particularly coincide with low ecosystem productivity in Italy and France, the Balkan and the Iberian Peninsula.

The number of events per month is not equally distributed throughout the year. There is a decline from June to November with a minimum usually around September in which only few events are detected. This reflects the time span of the dormant season since these months are usually too dry for ecosystem activity. There are some notable exceptions for land covers involving trees ("Forest (broadleaved)", "Forest (needleleaved)", "Mixed" and "Crops (Irrigated)") (see Fig. 5), as well as the most northern subregions of the Mediterranean, Italy and France and the Balkan Peninsula (see Fig. 6), where the number only decreases slightly during this period. These land cover classes and subregions are less affected by the characteristic dry period in summer. Forests have better access to soil moisture because they develop deeper roots (Bréda et al., 2006; Zhang et al., 2016), whereas irrigated areas obviously have an external water supply. The northern subregions are also moister than the southern Mediterranean.

Satellite-derived soil moisture data sets are prone to uncertainty, even though there have been considerable improvements in the last years (Gruber et al., 2019). Therefore, ecosystem vulnerability was also assessed for all land cover classes and subregions using soil moisture layers at 0–7 cm, 7–28 cm and 28–100 cm depth from the ERA5 Land reanalysis data set and compared to results obtained from the ESA CCI soil moisture product to verify the robustness of our results and whether specific biases are apparent (see Appendix A). Furthermore, certain land cover classes and subregions encompass a relatively small subset of grid points and thus non-significant ecosystem vulnerability might be related to data scarcity in some of these cases.

The spatial patterns of ecosystem vulnerability are displayed for four exemplary months of the year (see Fig. 7), whereas all twelve months can be found in the Appendix (see Fig. B1). In March in most western Mediterranean regions, low FAPAR values are associated with cold and wet conditions (blue colouring), whereas in the eastern Mediterranean vulnerability to hot conditions (purple and orange colouring) is already emerging at this time of the year. In June, almost all regions are vulnerable

to hot conditions and often also to dry conditions (purple and orange colouring) with exceptions in the northernmost regions such as the French Riviera, as well as mountainous regions such as the Peloponnese in Greece and the High Atlas in central Morocco. In September, there are often no low FAPAR anomalies occurring (black colouring) particularly in southern and inland regions, which are the hottest regions of the Mediterranean. The reason for this is that this time usually corresponds to the dormant season in these areas. In regions where events are detected during this time of the year, vulnerability to cold and dry

conditions (green colouring) prevails in most of the Mediterranean. In December, in most areas in the central Mediterranean, low FAPAR values coincide with cold and wet conditions (blue colouring), whereas in central Turkey and the southern Iberian Peninsula vulnerability to hot conditions (purple and orange colouring) occurs. It is noteworthy that for a given grid point at a given month, only 21 observations are available. Therefore, the robustness of the magnitude of ecosystem vulnerability of individual grid points is limited and should thus be interpreted with care. The maps in Figs. 7 and B1 primarily aim to identify

large-scale spatial patterns, but do not provide information on statistical significance at a grid point scale.

## 4 Discussion

### 4.1 Interpretation of temporal and spatial patterns in the Mediterranean

Our findings are in accordance with the characteristics of the Mediterranean climate regime, which is primarily energy-limited during winter and soil moisture-limited during summer (Schwingshackl et al., 2017). The vulnerability analysis allows a more

detailed investigation of the changes in ecosystem vulnerability to soil moisture and temperature throughout the course of the year for different land cover classes and subregions. In a wet regime, ecosystem activity is energy-limited, depending primarily on temperature and radiation, whereas in a transitional or dry system, soil moisture content is reduced and thus ecosystem activity is water-limited (Seneviratne et al., 2010; Zscheischler et al., 2015). From May to July, the Mediterranean is often vulnerable to hot and dry conditions, which is a typical feature of a soil moisture-limited regime (Seneviratne et al.,

2010). Heat waves are a frequent characteristic of the Mediterranean summer (Conte et al., 2002) and are often connected to persistent anti-cyclonic regimes and droughts (Mueller and Seneviratne, 2012; Ulbrich et al., 2012). The vulnerability to dry conditions in autumn indicates that moisture reservoirs are often still depleted after the summer, impairing the onset of the next vegetation cycle. By contrast, plant growth is inhibited by too low temperatures in autumn, which distinguishes it from the antecedent summer period. The general transition to vulnerability to cold conditions already in August is astonishing.

However, it should be noted that especially for the warmer regions – e.g. northwestern Africa, Turkey and the interior of Spain – either vulnerability to hot conditions prevails or no FAPAR anomalies are detected during this time (see Figs. 6 and B1), because August is outside of the growing season and the FAPAR values are usually at their annual minimum at this time of the year. During the phase of the water-limited regime, soil moisture depletion in combination with high atmospheric evaporative

demand leads to plant water stress and can ultimately cause plant mortality due to hydraulic failure or carbon starvation (van der Molen et al., 2011; Vicente-Serrano et al., 2020). As a coping strategy, plants e.g. reduce stomatal conductance to avoid hydraulic failure due to water loss by leaf transpiration, which consequently leads to reduced carbon uptake and thus decreased photosynthetic activity (van der Molen et al., 2011; Reichstein et al., 2013; Piao et al., 2019; Vicente-Serrano et al., 2020). The vulnerability to cold conditions in most months from November to April confirms that ecosystems are energy-limited in this period and is probably related to frost damage during cold spells. Related to the Cyprus Low, cold spells often co-occur with heavy precipitation in the eastern Mediterranean during this time (de Luca et al., 2020). Presumably, wet conditions only coincide with cold conditions, but are not damaging ecosystems as such. However, vulnerability of crops to wet conditions in winter was e.g. observed on the Iberian Peninsula in a study by Páscoa et al. (2017). While ecosystem activity in the northern Mediterranean is low during winter, this does not hold true for the southern Mediterranean – e.g. for some regions in Tunisia the NDVI peaks as early as December (Le Page and Zribi, 2019). Cloudiness during precipitation leads to reduced solar radiation and consequently lower surface temperature (Berg et al., 2015). This way, cold and wet conditions can lead to low transpiration rates of plants accompanied by low photosynthetic activity, leading to reduced extraction of soil moisture during that time period (Zscheischler et al., 2015). This highlights the bidirectional relation between vegetation and soil moisture, i.e. not only the state of the vegetation is dependent on soil moisture, but also vice versa. This mutual linkage is neglected in many studies (Dorigo et al., 2017).

Energy-limited regimes merge gradually into water-limited regimes from Scandinavia southwards to the Mediterranean in Europe (Teuling et al., 2009). Karnieli et al. (2019) investigated the relationship of the NDVI and land surface temperature at European scale, hypothesizing that a positive relationship indicates an energy-limited condition and a negative one a water-limited condition. Our results are mostly in agreement with the findings of their study that temperature and the NDVI are comprehensively negatively related in summer in Mediterranean Europe, whereas in spring this is only the case at the south-ernmost regions of Mediterranean Europe, while in other areas either neutral or negative relationships prevail. According to Le Page and Zribi (2019), temperature and the NDVI are always negatively correlated in northwestern Africa, while soil moisture and the NDVI are positively correlated. This indicates that this region is soil moisture-limited year-round, which is in good agreement with our results obtained using the ESA CCI soil moisture data set. However, the ERA5 Land soil moisture data set exhibits vulnerability to wet conditions in northwestern Africa in several months of the year, which might indicate lower suitability of this reanalysis data set to represent the soil moisture conditions in this region (see Figs. 6 and A2).

Extreme ecosystem impacts are not always connected to climatic extremes, but can also be caused by a combination of con-current moderate climatic drivers (Pan et al., 2020; van der Wiel et al., 2020). Furthermore, extreme ecosystem impacts are not solely related to soil moisture and temperature anomalies. Other potential causes are e.g. windthrow, pest outbreaks and fires, which often exhibit synergistic effects in combination with droughts and heat waves (Gouveia et al., 2012; Reichstein et al., 2013; Batllori et al., 2017; Ruffault et al., 2018). Furthermore, many ecosystems are managed, which also affects ecosystem productivity (Smit et al., 2008). These additional drivers should be taken into consideration when interpreting the results of this study.

The impact of climate extremes on ecosystems depends highly on their timing (Smith, 2011; Wolf et al., 2016; Piao et al., 2019). The sensitivity to heat varies with phenophase (Hatfield and Prueger, 2015) and the effect on the carbon cycle can
differ seasonally. High temperatures might e.g. increase carbon uptake by advancing spring onset, but may lead to uptake reductions in summer (Piao et al., 2019). In the same way, droughts can either accelerate the phenological cycle or inhibit plant productivity and their impact on vegetation is strongly connected to the seasonal variations of the water balance (Spano et al., 2013; Gouveia et al., 2017). The highest detrimental impacts on ecosystems by droughts in the Mediterranean have been reported at the beginning of the year at the peak of the growing season (Ivits et al., 2016; Peña-Gallardo et al., 2019).
The drought and heat wave in 2003 was comparably not that harmful to Mediterranean ecosystems, as it occurred in August, which is outside the main growing season (Ivits et al., 2016). The approach presented in this study helps to gain a better understanding of which stages of the year are vulnerable to which climatic condition. To our knowledge, none of the previous studies, which applied the framework for ecosystem vulnerability, accounted for the effects of seasonality so far. However, ecosystem responses are highly sensitive to the timing of events, therefore, it is crucial to consider this.

Climate change leads to seasonal shifts, which already becomes apparent in the strong phenological changes in the Mediterranean (Menzel et al., 2006; Gordo and Sanz, 2009, 2010). For example, higher temperatures lead to increased ecosystem productivity and subsequently higher evapotranspiration earlier in the growing season. Due to this, soil moisture is depleted faster and therefore more energy is transferred into sensible heat instead of latent heat. As a consequence of these hot and dry conditions, the growing season might end prematurely (Seneviratne et al., 2010; Lian et al., 2020). The time series used here
encompasses 21 years and is thus still too short for analyzing long-term trends. Nevertheless, our approach can potentially be used to monitor how vulnerability changes in future for all twelve months of the year by comparing vulnerability during different multi-year time spans if time series of sufficient length are available. Hot and dry days are getting more persistent in summer and unprecedented heat waves associated with Saharan warm air intrusions have occurred within the last years (Sousa et al., 2019; de Luca et al., 2020). Nevertheless, droughts and warm spells are increasing in spring as well (Vogel et al.,
2021), which can have detrimental implications for the Mediterranean ecosystems as spring is the main growing season. With temperature increases in future, vulnerability to cold conditions might be constrained to a shorter time frame, whereas the time span with vulnerability to hot conditions might expand within the year. Increasing aridity is projected in the Mediterranean, especially during winter and spring (Samaniego et al., 2018), while at the same time heavy precipitation events are projected to increase (Toreti and Naveau, 2015). Thus, it remains difficult to determine how vulnerability to dry and wet conditions will
evolve in future.

## 4.2 Potential limitations of the methodological procedure

The presented method depends heavily on the quality of the employed data types for both the two drivers and the impact proxy. Several limitations regarding moisture data are well-known, e.g. the coarse spatial resolution impairs assessments at local scales. Furthermore, satellite-based soil moisture is limited to the retrieval of surface soil moisture, while deeper-reaching
root-zone soil moisture is the actual ecologically relevant variable. Satellite-based soil moisture is only representative for the first five centimetres of the soil layer. The root zone of plants is usually deeper, which reduces the explanatory power of

satellite-based soil moisture for drought impacts on ecosystems (Liu et al., 2016; Dorigo et al., 2017; West et al., 2019). For example, soil drying during summer affects primarily the top soil layer, while drying in deeper layers shows a lagged response, because upward capillary flow from these layers is comparatively slow (Berg et al., 2017). Nicolai-Shaw et al. (2017) found

that soil moisture data from ESA CCI was a good indicator for drought in grasslands, while forests exhibited weaker responses, probably due to access to deeper soil layers for forests compared to grasslands. However, we also assessed vulnerability to soil moisture at the depths 0–7 cm, 7–28 cm and 28–100 cm using reanalysis data from ERA5 Land and the patterns obtained at the deeper layers 7–28 cm and 28–100 cm are in large part similar to the ones of the layer at 0–7 cm (see Appendix A). This indicates that the assessment of the top soil layer is able to yield results, which are valid for a larger proportion of the soil

column. Coupling of land surface models with satellite-based surface soil moisture can further enhance knowledge on the status of root-zone soil moisture in future (Dorigo et al., 2017; Tramblay et al., 2020). Futhermore, it should be noted that validations of the ESA CCI soil moisture data set with in situ observations from Mediterranean sites in Spain, France and Turkey showed high agreement (Albergel et al., 2013; Dorigo et al., 2015; Bulut et al., 2019). Also the FAPAR product from the Copernicus Climate Change Service has been validated with observation data from Tunisia, Italy, Spain and France, primarily for a variety

of crop types, as well as a deciduous broadleaf forest in Italy and a needle-leaf forest in Spain (Fuster et al., 2020). The FAPAR is often assumed to be directly linked to productivity. However, droughts might lead to physiological changes such as stomata closure, which are not apparent in the spectral characteristics of the canopy and thus in the FAPAR but nevertheless invoke a decreased productivity. This was e.g. the case in forest ecosystems during the drought and heat wave event in 2003 in Europe (Reichstein et al., 2007; Zhang et al., 2016).

The Mediterranean Basin is characterised by large spatial variability because of its complex topography (Lionello et al., 2006). The relatively coarse resolution of the ESA CCI soil moisture data set is currently limiting the representation of this high spatial complexity (Crocetti et al., 2020). Many land cover classes express similar patterns over the course of the year according to our results. This could potentially indicate that grid points are sometimes not homogeneous enough, but rather represent a mixture of several land cover classes due to the coarse resolution of 0.25 °. The ESA CCI land cover product applied in this

study is a state of the art data set; a more detailed data set is currently not available for the Mediterranean Basin as a whole. The ESA CCI land cover classification allows only for the differentiation of major plant functional types and future studies might benefit from a more refined land cover classification scheme with a broader variety of land cover classes. Furthermore, the subregions used in this study are not fully homogeneous and there is a certain variability within a given subregion. Thus, the patterns identified in this study (see Figs. 5 and 6) cannot always be inferred for an entire subregion. Therefore, the ecosystem

vulnerability maps (Figs. 7 and B1) should be additionally examined for the identification of potentially deviating patterns within subregions.

Many studies do not consider lagged effects in their design and the choice of a suitable time scale to account for such effects is not trivial and under debate (Zeng et al., 2013; Ivits et al., 2016). Response time varies depending on the type of event and the affected ecosystem. The response lag of vegetation is land cover-specific, as plants have various regulatory physiological

functions to react to changes in soil moisture such as stress memory, water storage and stabilisation activities at the community level (van der Molen et al., 2011; Niu et al., 2014; Zhang et al., 2017). Faster response times to droughts are observed for pasture

and crops compared to shrubs and forests (Chen et al., 2014; Bachmair et al., 2018). Generally, responses to drought are slower in semi-arid and sub-humid biomes compared to arid biomes (Vicente-Serrano et al., 2013). A study by Ivits et al. (2016) at European scale found that vegetation in the Mediterranean responds slowly to meteorological droughts compared to most other European regions. Impacts on vegetation by meteorological and soil moisture droughts are often largest within the preceding one to two months (Zeng et al., 2013; Chen et al., 2014; Wu et al., 2015; Papagiannopoulou et al., 2017; Bachmair et al., 2018), which is the reason we decided on a three-month time scale in the moving average applied to the environmental drivers in our approach. Temperature responses are usually faster than responses to drought, but can still exhibit lagged responses up to a few months (Zeng et al., 2013; Papagiannopoulou et al., 2017). Temperature and soil moisture anomalies are usually analysed on different time scales (typically on a daily scale for temperature and on a monthly scale for soil moisture), which renders their joint assessment difficult. Ecosystem impacts can also vary substantially on a temporal scale from e.g. temporary changes in productivity to persistent regime shifts (Crausbay et al., 2017). Therefore, using a single time scale might not capture all relevant temporal dynamics. The choice of the optimal time scale is non-trivial and e.g. time scales of less than a month for investigating drought impacts on vegetation have also been suggested (West et al., 2019).

Our analysis is year-round without being explicitly restricted to the months of the growing season, which makes it easily transferable to any study area. We decided this for two reasons. First, it is complex to account only for the months of the growing season, as there is a large variability depending on latitude and longitude within the Mediterranean Basin (Lionello et al., 2006). Second, the analysis is implicitly limited to the growing season, because FAPAR deviations during the dormant season are expected to be small and thus will exceed the extremeness threshold only on rare occasions. In our study, it can be clearly noted that the number of detected events is not distributed equally throughout the course of the year. They are at a minimum at the transition from summer to autumn when ecosystem activity is low in the Mediterranean (see section 3.2). Therefore, large areas – especially in the interior of the countries – are under-represented in these months. Results for months during the dormant season should be interpreted cautiously (Ivits et al., 2016), taking into account that they depend on a considerably lower number of events. These events might be representative solely for specific ecosystems that are still active at this time of the year or may partially result from noise in the data.

## 5 Conclusions

The seasonal ecosystem vulnerability analysis presented in this study helps identifying at which time of the year vulnerability to a certain climatic condition occurs. The vulnerability of Mediterranean ecosystems to the concurrent climatic drivers temperature and soil moisture was successfully assessed using the FAPAR as a proxy for ecosystem productivity, with a focus on the variation of impacts with seasonality. Our results are in line with the characteristic intra-annual change between an energy-limited and a water-limited regime from winter to summer in the Mediterranean (Schwingshackl et al., 2017). In general, three seasonal stages of vulnerability are identified throughout the year: 1) vulnerability to hot and dry conditions in late spring to midsummer, 2) vulnerability to cold and dry conditions from the end of summer to mid-autumn and 3) vulnerability during cold and wet conditions from the end of autumn to mid-spring. There are several regions which deviate from this pattern, e.g. the

hot and dry regime is extended from spring to autumn in Turkey, whereas the Balkan Peninsula is continuously energy-limited throughout the year and not vulnerable to hot conditions. Our results point out the necessity to incorporate seasonality in the vulnerability analysis concept, as well as to examine vulnerability at a subregional scale to account for the large spatial and temporal variability in the Mediterranean. Increasing aridity and fast changes in the phenological cycle are observed in the Mediterranean Basin due to climate change (Gao and Giorgi, 2008; Gordo and Sanz, 2010). The approach for detecting sea-
sonal ecosystem vulnerability opens novel opportunities for developing early-warning tools to identify detrimental ecosystem conditions, water limitations and irrigation demand in near real time and for performing long-term assessments of ecosystem vulnerability and change for the near- and mid-future climate scenarios.

*Code and data availability.* The code can be retrieved from https://gitup.uni-potsdam.de/joschavogel/ecosystem_vulnerability. All data sets used in this study are publicly available.

**Appendix A: Comparison of ecosystem vulnerability using soil moisture from ESA CCI and ERA5 Land**

The ERA5 Land soil moisture layer at 0–7 cm gives very similar results compared to the ESA CCI data set in the second half of the year (August–December) for most land cover classes (see Fig. A1), where the patterns are identical in most cases – for "All land cover classes" they are in agreement from June to December. However, in spring they often deviate, e.g. in May where dry conditions arise in the ERA5 Land data set, whereas using ESA CCI there is no significant vulnerability to dry conditions
for many land cover classes. For land cover classes such as "Crops (rainfed)" vulnerability to dry conditions in May seems realistic, as various crops are prone to drought in their reproductive phase (Zhang and Oweis, 1999; Daryanto et al., 2016), which indicates that ERA5 Land might give more plausible results for the month of May. "Shrubland" is often prone to dry conditions in the second half of the year in the ESA CCI data set, whereas according to the ERA5 Land data set it is not. Also "Forest (broadleaved)" is prone to dry conditions from June to October in the ESA CCI data set, unlike in the ERA5 Land data
set where it is vulnerable to dry conditions from September to October, but not during summer. However, there is no apparent systematic bias over all classes, rather it changes by month. So, in February, vulnerability in the ERA5 Land data set is e.g. leaning more towards dry conditions, whereas in July this pattern is reversed.

During most of the year, the majority of subregions coincide well in both data sets, but there are exceptions (see Fig. A2). There is vulnerability to dry conditions in August in the Balkan, the Iberian Peninsula and northwestern Africa for ESA
CCI soil moisture, whereas for ERA5 Land this is reversed or insignificant. For northwestern Africa, ERA5 Land detects lower vulnerability to dry conditions than ESA CCI throughout the course of the year. In addition, in the Iberian Peninsula vulnerability to wet conditions is pronounced at the beginning of the year for ESA CCI, whereas for ERA5 Land most months during this period show vulnerability to dry conditions.

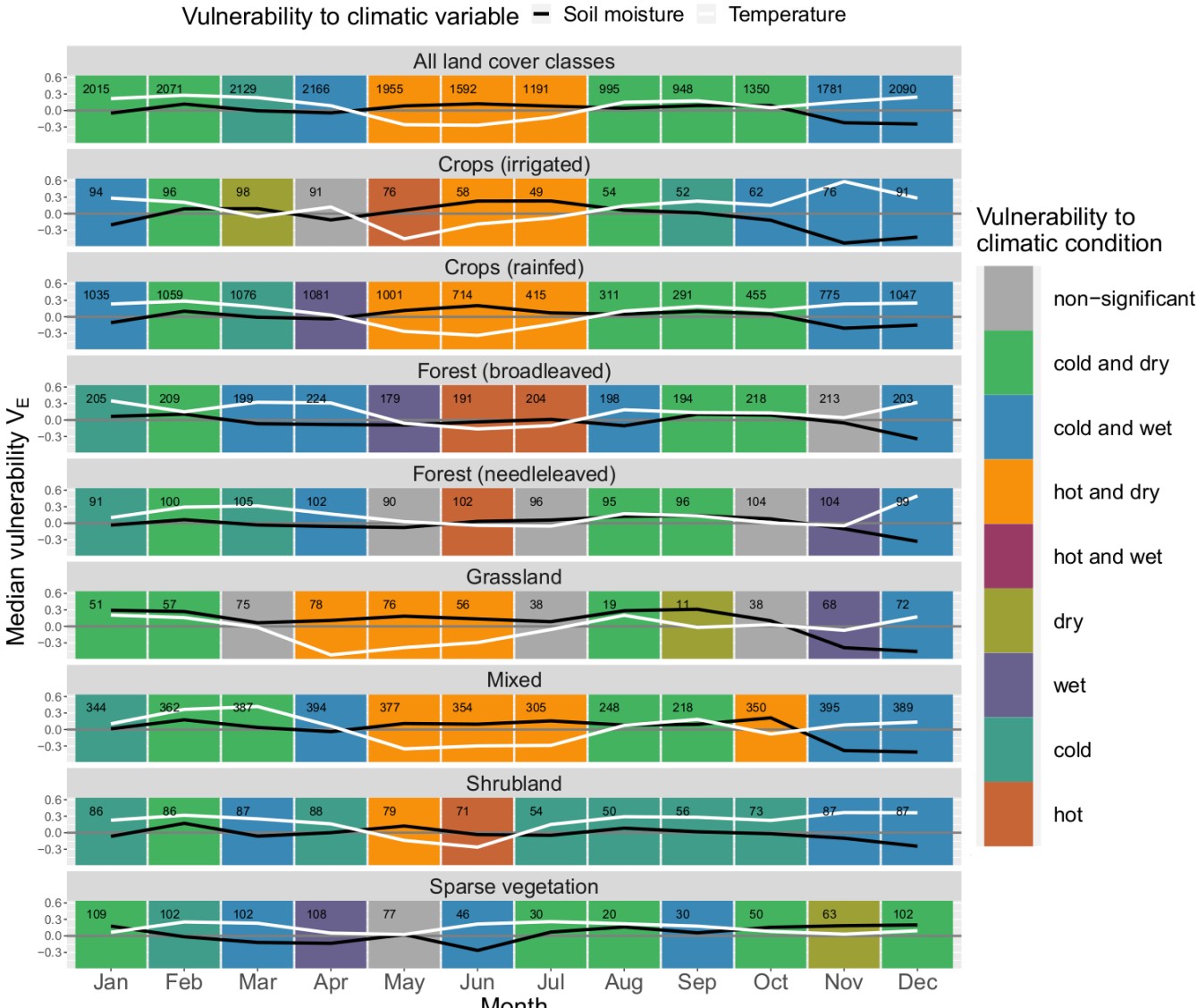

**Figure A1.** Median monthly ecosystem vulnerability per land cover. Vulnerability to temperature (ERA5 Land) is shown in white and vulnerability to soil moisture (ERA5 Land at depth 0–7 cm) is shown in black for each month of the year (columns) for each land cover (rows). Months with statistically significant deviation of climatic drivers during non-hazardous and hazardous ecosystems conditions according to the Mann-Whitney U test based on a significance level $\alpha = 0.05$ are shown in colour (see legend), all other months are shown in grey. The number of grid points in which an event has occurred in this month and land cover within the period 1999-2019 is shown in the upper left corner of each panel.

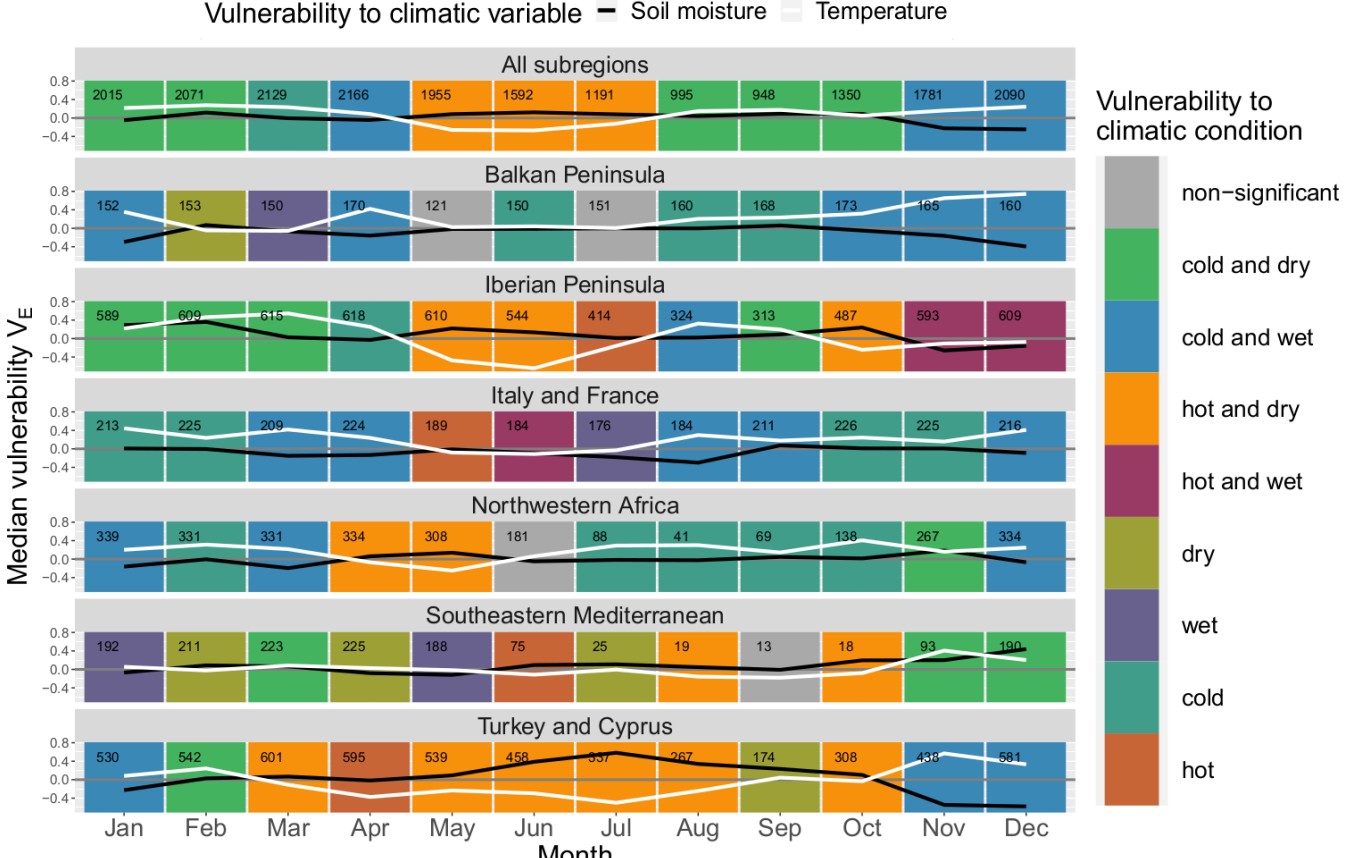

**Figure A2.** Median monthly ecosystem vulnerability per subregion. Vulnerability to temperature (ERA5 Land) is shown in white and vulnerability to soil moisture (ERA5 Land at depth 0–7 cm) is shown in black for each month of the year (columns) for each land cover (rows). Months with statistically significant deviation of climatic drivers during non-hazardous and hazardous ecosystems conditions according to the Mann-Whitney U test based on a significance level $\alpha = 0.05$ are shown in colour (see legend), all other months are shown in grey. The number of grid points in which an event has occurred in this month and land cover within the period 1999-2019 is shown in the upper left corner of each panel.

In addition to the soil moisture layer corresponding to 0–7 cm soil depth, vulnerability to soil moisture was also analysed for the layers at 7–28 cm and 28–100 cm (see Figs. A3, A4, A5, A6). The patterns at these deeper layers largely coincide with the surface soil moisture layer (see Figs. A1, A2).

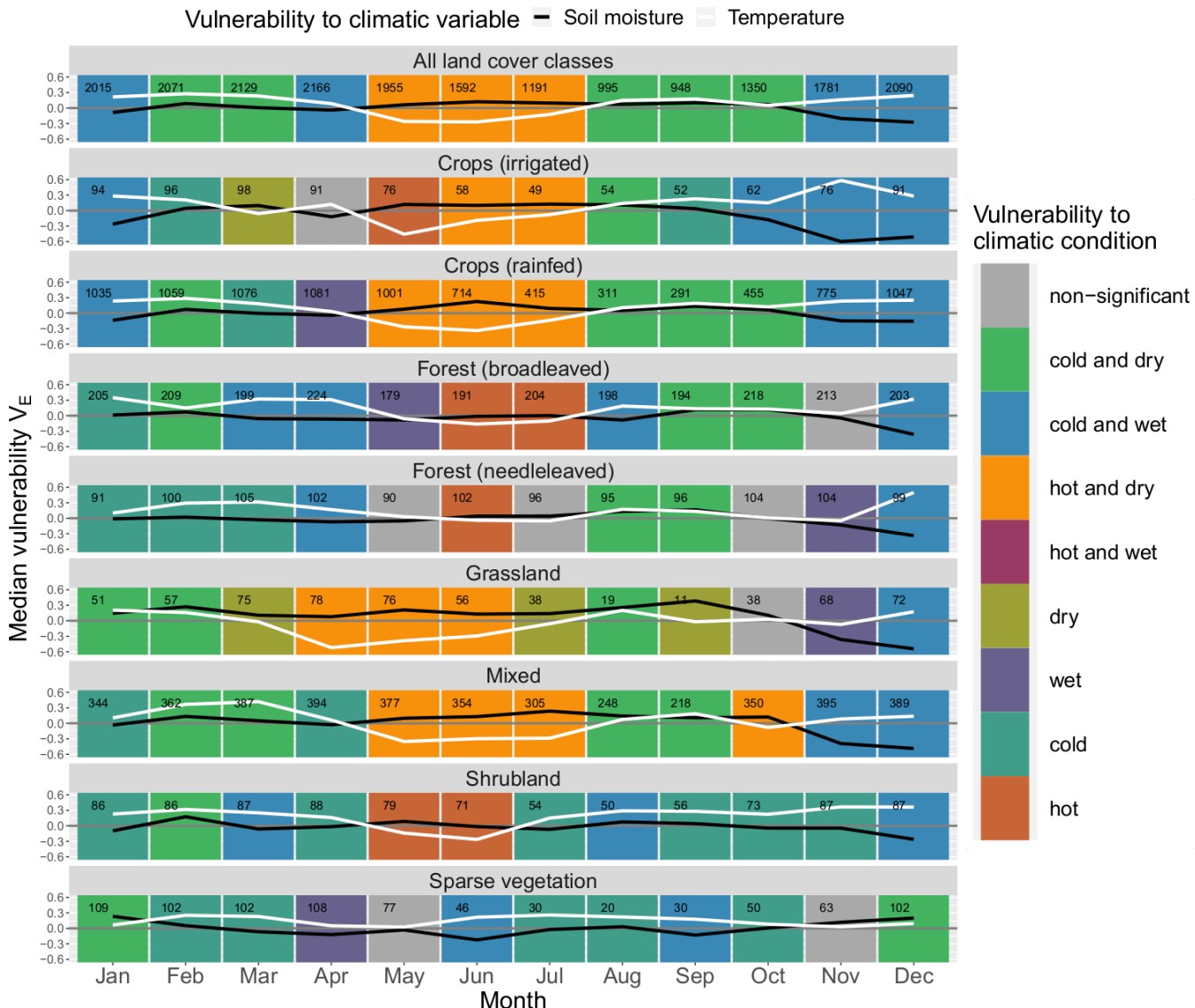

**Figure A3.** Median monthly ecosystem vulnerability per land cover. Vulnerability to temperature (ERA5 Land) is shown in white and vulnerability to soil moisture (ERA5 Land at depth 7–28 cm) is shown in black for each month of the year (columns) for each land cover (rows). Months with statistically significant deviation of climatic drivers during non-hazardous and hazardous ecosystems conditions according to the Mann-Whitney U test based on a significance level $\alpha = 0.05$ are shown in colour (see legend), all other months are shown in grey. The number of grid points in which an event has occurred in this month and land cover within the period 1999-2019 is shown in the upper left corner of each panel.

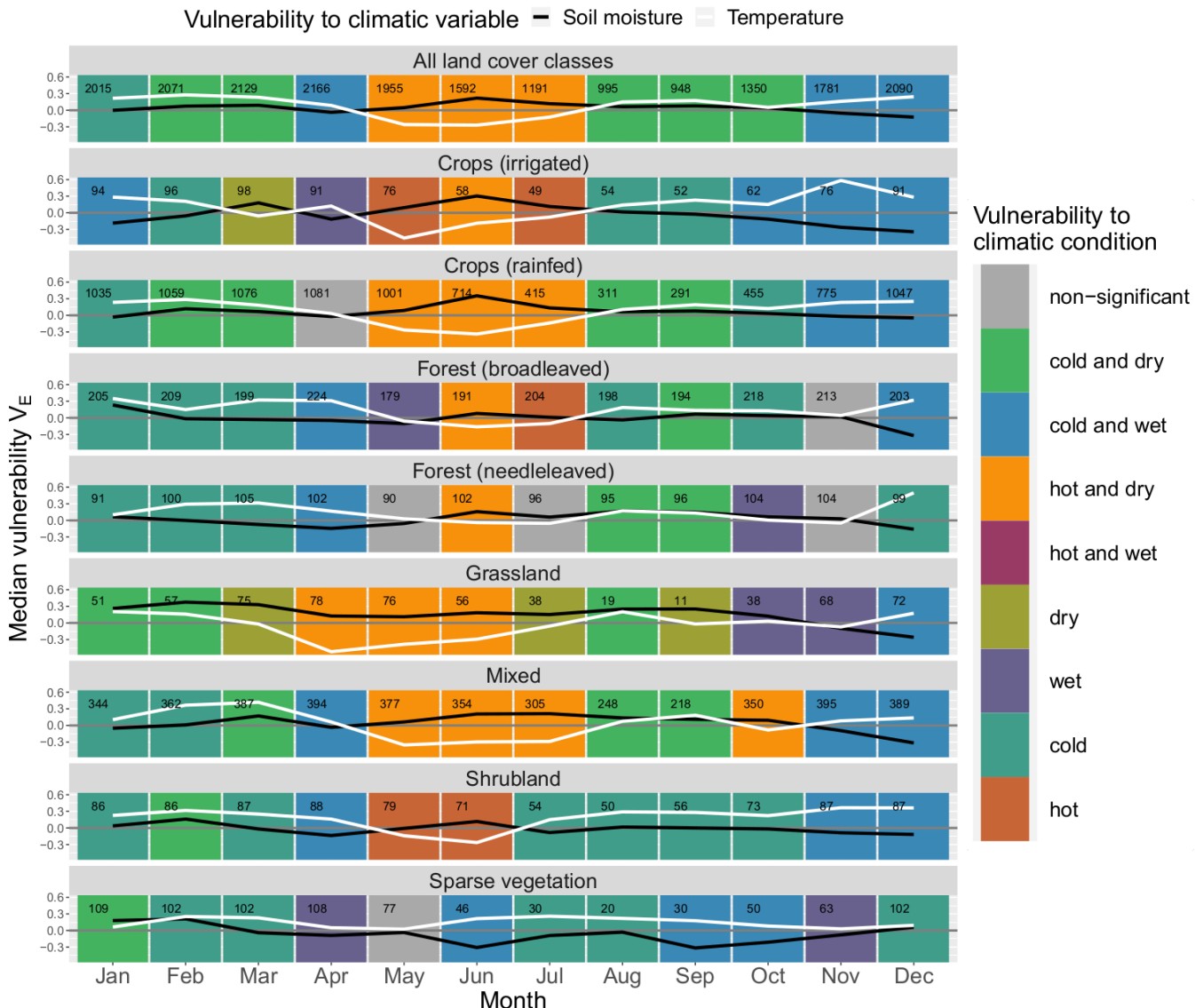

**Figure A4.** Median monthly ecosystem vulnerability per land cover. Vulnerability to temperature (ERA5 Land) is shown in white and vulnerability to soil moisture (ERA5 Land at depth 28–100 cm) is shown in black for each month of the year (columns) for each land cover (rows). Months with statistically significant deviation of climatic drivers during non-hazardous and hazardous ecosystems conditions according to the Mann-Whitney U test based on a significance level $\alpha = 0.05$ are shown in colour (see legend), all other months are shown in grey. The number of grid points in which an event has occurred in this month and land cover within the period 1999-2019 is shown in the upper left corner of each panel.

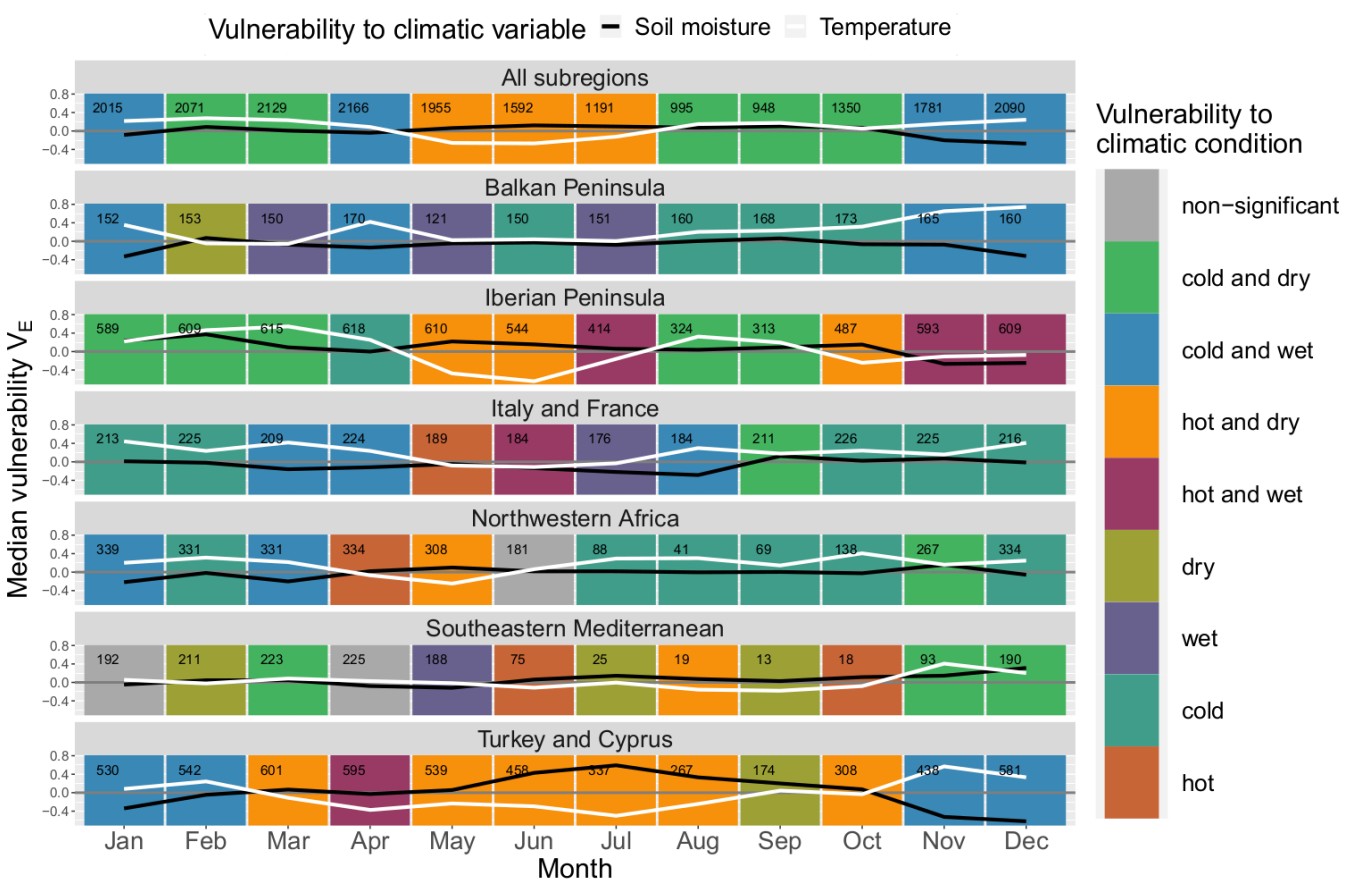

**Figure A5.** Median monthly ecosystem vulnerability per subregion. Vulnerability to temperature (ERA5 Land) is shown in white and vulnerability to soil moisture (ERA5 Land at depth 7–28 cm) is shown in black for each month of the year (columns) for each land cover (rows). Months with statistically significant deviation of climatic drivers during non-hazardous and hazardous ecosystems conditions according to the Mann-Whitney U test based on a significance level $\alpha = 0.05$ are shown in colour (see legend), all other months are shown in grey. The number of grid points in which an event has occurred in this month and land cover within the period 1999-2019 is shown in the upper left corner of each panel.

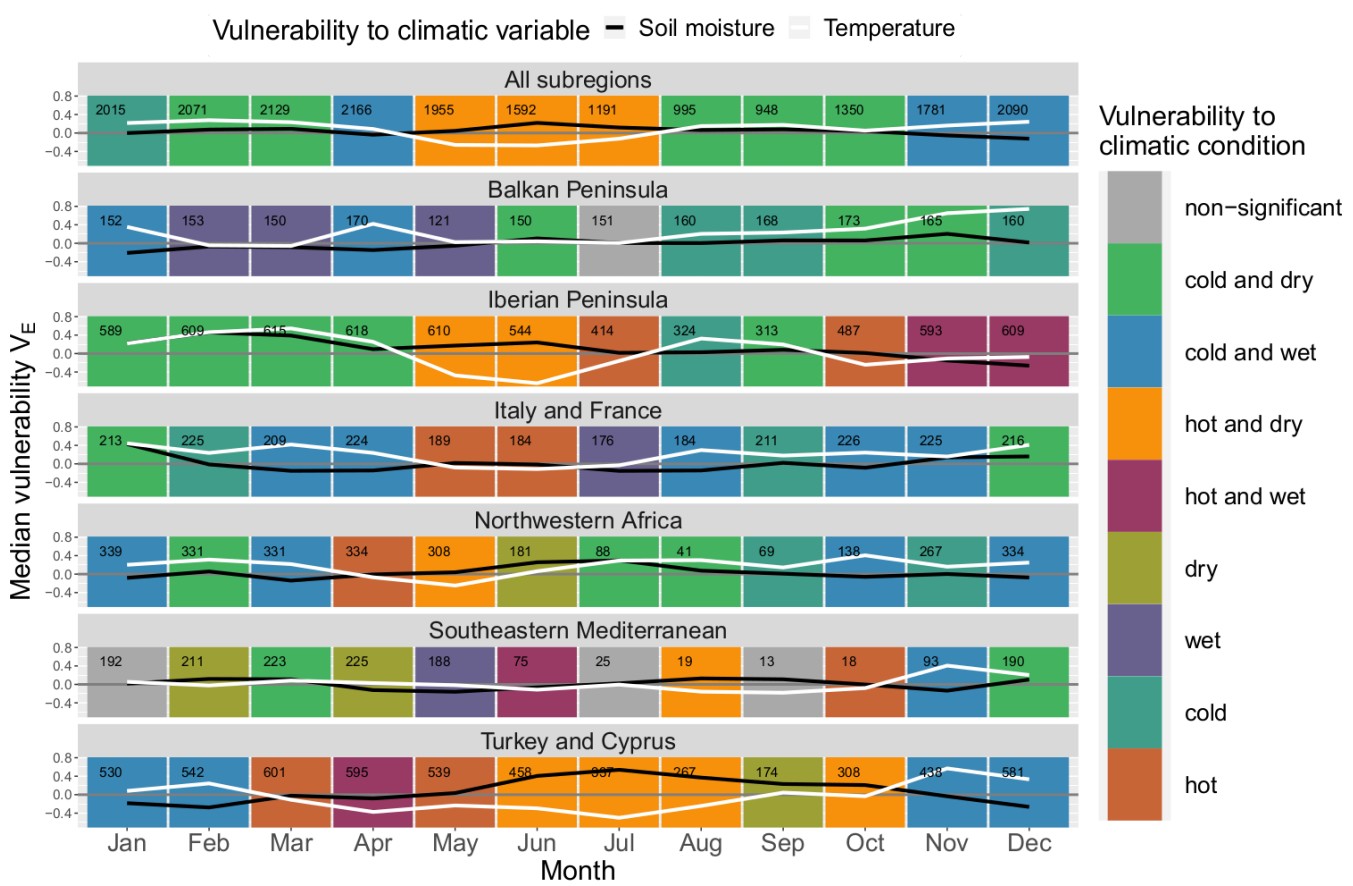

**Figure A6.** Median monthly ecosystem vulnerability per subregion. Vulnerability to temperature (ERA5 Land) is shown in white and vulnerability to soil moisture (ERA5 Land at depth 28–100 cm) is shown in black for each month of the year (columns) for each land cover (rows). Months with statistically significant deviation of climatic drivers during non-hazardous and hazardous ecosystems conditions according to the Mann-Whitney U test based on a significance level $\alpha = 0.05$ are shown in colour (see legend), all other months are shown in grey. The number of grid points in which an event has occurred in this month and land cover within the period 1999-2019 is shown in the upper left corner of each panel.

**Appendix B:  Further materials**

**Table B1.** Overview of the land cover classes aggregation.

| Number | Original class | Aggregated class |
|---|---|---|
| 10 | Cropland, rainfed | Crops (rainfed) |
| 11 | Cropland, rainfed, herbaceous cover | |
| 12 | Cropland, rainfed, tree or shrub cover | |
| 20 | Cropland, irrigated or postflooding | Crops (irrigated) |
| 30 | Mosaic cropland (>50%) / natural vegetation (tree, shrub, herbaceous cover) (<50%) | Mixed |
| 40 | Mosaic natural vegetation (tree, shrub, herbaceous cover) (>50%) / cropland (<50%) | |
| 100 | Mosaic tree and shrub (>50%) / herbaceous cover (<50%) | |
| 60 | Tree cover, broadleaved, deciduous, closed to open (>15%) | Forest (broadleaved) |
| 62 | Tree cover, broadleaved, deciduous, open (15-40%) | |
| 70 | Tree cover, needleleaved, evergreen, closed to open (>15%) | Forest (needleleaved) |
| 120 | Shrubland | Shrubland |
| 130 | Grassland | Grassland |
| 150 | Sparse vegetation (tree, shrub, herbaceous cover) (<15%) | Sparse vegetation |
| 153 | Sparse herbaceous cover (<15%) | |
| 200 | Bare areas | |
| 190 | Urban areas | None (Omitted) |
| 210 | Water bodies | None (Omitted) |

**Table B2.** Overview of the six subregions and the corresponding countries used in this study

| Short Name | Long Name | Countries |
|---|---|---|
| IBE | Iberian Peninsula | Portugal, Spain |
| IAF | Italy and France | France, Italy |
| BAL | Balkan Peninsula | Albania, Bosnia and Herzegovina, Bulgaria Croatia, Greece, North Macedonia, Montenegro |
| TAC | Turkey and Cyprus | Cyprus, Turkey |
| SEM | Southeastern Mediterranean | Iran, Iraq, Israel and Palestinian territories Jordan, Lebanon, Libya, Syria |
| NWA | Northwestern Africa | Algeria, Morocco, Tunesia |

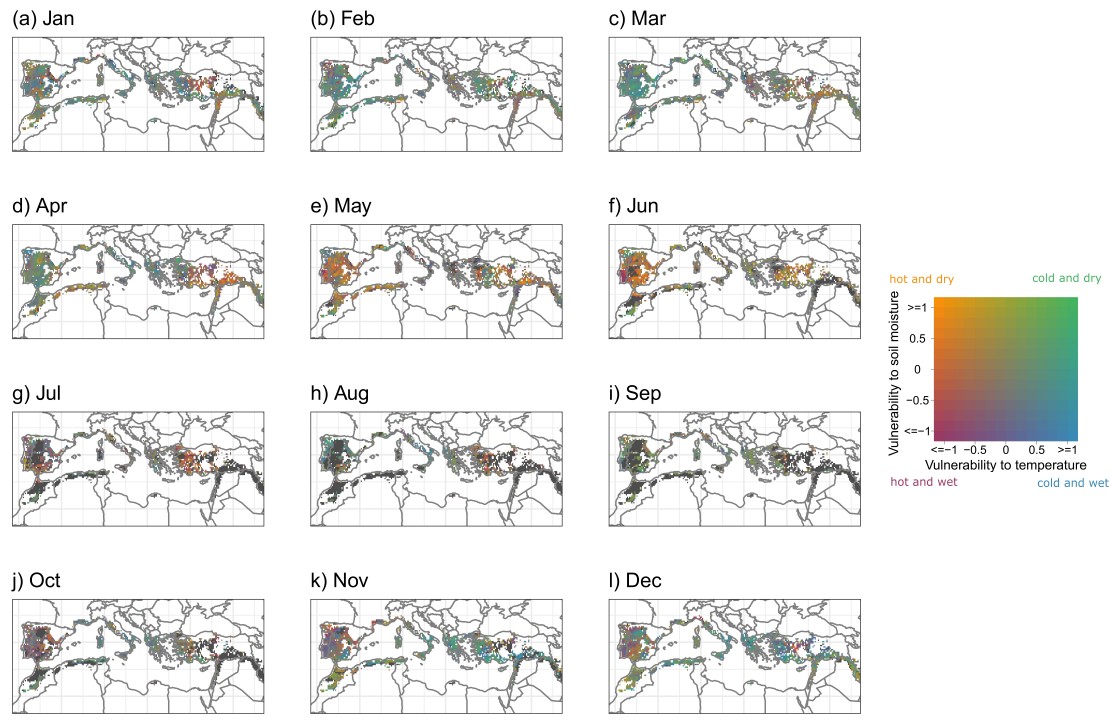

**Figure B1.** Average monthly vulnerability to soil moisture (ESA CCI) and temperature (ERA5 Land) in the Mediterranean Basin for (a) January, (b) February, (c) March, (d) April, (e) May, (f) June, (g) July, (h) August, (i) September, (j) October, (k) November and (l) December. Grid points without any events during the respective month are displayed black.

*Author contributions.* JV, EP and VA designed the study and the methodology. JV developed the computer code, performed the analysis and visualised the results. EP supervised the research project. JV wrote the original draft with contributions from all co-authors.

*Competing interests.* The authors declare that they have no conflict of interest.

*Acknowledgements.* We acknowledge the support of the DFG research training group "Natural Hazards and Risks in a Changing World" (NatRiskChange GRK 2043) and Open Access Publishing Fund of University of Potsdam. We would like to thank the guest editor Bart van den Hurk, the Co-Editor-in-Chief Michael Bahn, as well as Niko Wanders and the other two anonymous reviewers for their valuable feedback to the manuscript.

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
