# Peer review of "Seasonal ecosystem vulnerability to climatic anomalies in the Mediterranean"

_Biogeosciences, 2021_

## Author Comment (AC1)

**Answer to Anonymous Referee #1**

We would like to thank the anonymous reviewer for the comments.

The reviewer raised concerns regarding a) the quality of soil moisture data, b) the spatial aggregation of land cover classes and Mediterranean subregions and c) the standardization technique.

The satellite-based soil moisture data set from ESA CCI is a state of the art data set, which underwent validation and quality assessment and has been used frequently for similar purposes in the scientific literature. Furthermore, the results from the reanalysis soil moisture data set from ERA5 Land are in agreement with the ones from ESA CCI in general. For details, we refer to our answer regarding the comments on lines 45-54 / 75-80.

The broad aggregation of land cover classes is common at such large spatial scales and a significantly more detailed land cover product is not available for the entire Mediterranean Basin to our knowledge. We argue that the classification of Mediterranean subregions allows visualizing certain large-scale patterns, while more detailed at pixel scale can be retrieved from the spatial maps provided in the article. For details, we refer to our answers regarding the comments on lines 69, Figure 5 and Figure 6.

The z-score standardization technique used here is commonly applied and does not bias our results. We additionally performed the analysis by dividing by the interquartile range instead of the standard deviation, however differences in the results of both approaches are negligible. For details, we refer to our answer regarding the comment on lines 100-105.

A detailed answer to all comments follows below.

**Major comments**

**Lines 45-54 / 75-80:**

To address the uncertainties of the satellite-based soil moisture data set from ESA CCI, we additionally used the soil moisture reanalysis data set from ERA5 Land for the layers 0-7 cm, 7-28 cm and 28-100 cm (corresponding to the soil moisture data sets used in Nicolai-Shaw et al. 2017) to assess the soundness of our results. Both data sets show agreement in the overall patterns (see Appendix A). We only show the first layer (0-7 cm) from ERA5 Land in the submitted manuscript, because it is rather well correlated with deeper layers and there are only subtle changes in vulnerability between these layers. This is in line with findings by Orth et al. (2020) who also found only minor difference between surface and deep layers. In addition to Figure A1 and A2, we will add corresponding figures for the layers 7-28 cm and 28-100 cm, to enable the reader to assess how vulnerability to soil moisture varies within the top 100 cm of the soil according to the reanalysis data set from ERA5 Land. We appreciate that the additional usage of the soil moisture data set from ERA5 Land should be described more prominently in the manuscript (currently in lines 5-6, 80-81 and 200-202). We will further mention its usage in line 58 in the introduction.

The optimal hydrologic variable referring to ecosystem vulnerability is plant available water within the root zone. However, observations of this variable are not available for large spatial scales. Therefore, it is necessary to use either related observational data as proxies such as precipitation or satellite surface moisture data or data resulting from land surface models or assimilation of model and observations, all of which have specific constraints.

Soil moisture values are included in the land-data assimilation in the ERA5 Land reanalysis data set based on a point-wise Simplified Extended Kalman Filter (de Rosnay et al. 2013, Hersbach et al. 2020).

Surface soil moisture is linked to root-zone soil moisture via soil water redistribution to deeper layers and therefore remotely sensed surface soil moisture is used for the approximation of root zone soil moisture in land data assimilation systems (Maggioni and Houser 2017, Orth et al. 2020). We are well aware that the dynamics of surface soil moisture are directly linked to wetting and drying processes, i.e. rainwater infiltration and soil evaporation. Thus, soil water dynamics of deeper soil layers show a dampened and delayed dynamics compared to the surface layer, but still there is a significant correlation between them (Akbar et al. 2018). ERA5 is the first ECMWF reanalysis data set, which includes satellite information for the derivation of soil moisture estimates (Hersbach et al. 2020).

We will add a further sentence on the delayed connection of surface and deeper soil layers in line 297:

"For example, it takes some time for the propagation of surface drying to deeper soil layers, because of the slow capillary flow (Berg et al. 2017)."

We further argue that using satellite-based soil moisture data in large-scale ecological applications is a state-of-the art method. We want to emphasize that remote sensing soil moisture products have significantly improved over the last years (Mohanty et al. 2017, Gruber et al. 2019) (see lines 199-200), are increasingly being used to assess the impact of droughts on plant productivity (Dorigo and de Jeu 2016) and the ESA CCI soil moisture has been applied succesfully in a large number of studies for ecological applications. For an overview we refer e.g. to Table 5 and section 4.3 in Dorigo et al. (2017). This shows that surface soil moisture – despite its undisputed limitation – can give valuable insights on the state of the ecosystems. Denissen et al. 2019 state that satellite surface soil moisture is well suited to infer the state of the vegetation and corresponding land-atmosphere interactions during climate extremes. The transition from an energy- to a water-limited regime is marked by the critical soil moisture regions (Denissen et al. 2019), which makes soil moisture an appropriate variable especially in Mediterranean environemnets (Szczypta et al. 2014). Chen et al. (2014) assessed variability in the NDVI over Australia and conclude that not only precipitation but also remotely sensed soil moisture data from ESA CCI can be a good predictor for vegetation growth. A further example where ESA CCI soil moisture is used for biospheric drought effect assessment can be found in Orth et al. (2020) (published in Biogeosciences). Nicolai-Shaw et al. (2017) emphasize that remotely sensed soil moisture is a valuable addition or might even be able to replace other soil moisture proxies for the investigation of land-vegetation-atmosphere-dynamics (see lines 54-55). The authors highlight the usage of remote sensing based soil moisture for the assessment of drought development. They found strong responses of grasslands to soil moisture droughts, while forests showed weaker responses and relate this to the shallower rooting depth of grasslands compared to forests.

We will add a sentence on this at line 298:

"Nicolai-Shaw et al. (2017) found that soil moisture data from ESA CCI was a good indicator for drought in grasslands, while forests exhibited weaker responses, probably due to access to deeper soil layers for forests compared to grasslands."

For a comparison of the ESA CCI soil moisture data set with gridded precipitation, see e.g. Dorigo et al. (2012, 2017). Furthermore, the ESA CCI soil moisture data set was even used to create a global precipitation product, which was compared to state-of-the-art precipitation data sets and showed relatively good performance (Ciabatta et al. 2018). We also would like to emphasize that ESA CCI soil moisture has been validated succesfully with in situ observations e.g. in Spain and France at depths from 5 to 30 cm (see lines 299-301; Albergel et al. 2013, Dorigo et al. 2015) and Turkey (Bulut et al. 2019).

Finally, precipitation does not directly translate into plant available water within the root zone (de Boeck et al. 2011; see lines 47-49) and is thus not necessarily superior to surface soil moisture data for the assessment of plant available soil water. Several other processes play a role, such as evapotranspiration (especially important in the Mediterranean), runoff, topography, soil properties and irrigation (Mohanty et al. 2017).

**Line 69:**

A general remark on the data used in this study: There is a certain mismatch between the desirable spatial scale for assessing ecological impacts and the actually available climatic and land cover data sets (Ummenhofer et al. 2017). So, while remote sensing offers consistent large spatial and temporal coverage, there remains a trade-off regarding their coarse spatial resolution.

A more detailed land cover classification map would be beneficial, but is not available for the Mediterreanean Basin to our knowledge (CORINE Land Cover would show greater detail but is only available for the European side of Mediterranean basin).

We argue that it is common to use such rather broad categories, which rather resemble plant functional types than specific plant communities, for the analysis of the impact of climatic extremes on ecosystems. Plant functional types are partially based on climatic preferences (Bonan 2016). While there are certain simplifications, their distinctions nevertheless provide valuable insights on key ecological properties. For example, when comparing broadleaf and needleleaf forests, the former have high photosynthetic rates and stomatal conductance, the latter have lower photosynthetic rates and conductance (Bonan 2016), and it has been shown, that they show differences in drought-induced growth reductions leading to tree mortality (Cailleret et al. 2016). As a further example, Teuling et al. 2010 investigate the differing responses to heatwaves for forests and herbaceous perennial vegetation using data from various European flux tower sites. Furthermore, dynamic global vegetation models are usually based on plant functional types and these models are built with the intent to investigate the feedback of ecosystems and climate (Bonan 2016).

Therefore, we argue that we use an adequate state of the art classification scheme for investigating interactions of ecosystems and climate at such spatial scales for our type of research.

Land cover classification schemes like the one in our study are commonly applied; examples of studies using the same or similar land cover classifications in such a context include Ceccherini et al. (2014), Baumbach et al. (2017) (published in *Biogeosciences*), Nicolai-Shaw et al. (2017) and Buitenwerf et al. (2018).

We would further like to point out that we only include grid points belonging to the Köppen-Geiger classes Csa ("Warm temperate climate with dry and hot summer") or Csb ("Warm temperate climate with dry and warm summer"), i.e. areas with alpine grasslands and semiarid steppes are not considered in the study.

**Lines 100-105:**

Calculating z-scores is a commonly applied standardisation technique (Phillips 2018). It does not lead to a bias in the tails of the distribution, but merely rescales the data. The purpose of this transformation here is to rescale soil moisture and temperature to a common scale, so they can be analysed and displayed jointly on this standardised scale (while maintaining their original distribution shape) (Orth et al. 2020). For illustration, we show time series and histograms for an exemplary pixel for all three variables before and after the transformation below (Figs. R1 and R2).

Figure R1: Time series of temperature, soil moisture and FAPAR from 1999-2019 before (upper row) and after standardisation (lower row) for an exemplary pixel

---

## Author Comment (AC3)

**Answer to editor**

We would like to thank you for your valuable comments and the five points of criticism and your suggestions how to address them. We will discuss these points below: a) standardisation technique, b) maturity of remotely sensed soil moisture data, c) spatial aggregation of land cover data, d) ground truth and e) trend analysis (these answers are also part of the individual responses to the reviewers).

**a) Standardisation technique**

Calculating z-scores is a commonly applied standardisation technique (Phillips 2018). It does not lead to a bias in the tails of the distribution, but merely rescales the data. The purpose of this transformation here is to rescale soil moisture and temperature to a common scale, so they can be analysed and displayed jointly on this standardised scale (while maintaining their original distribution shape) (Orth et al. 2020). For illustration, we show time series and histograms for an exemplary pixel for all three variables before and after the transformation below (Figs. R1 and R2).

[Figure]

*Figure R1: Time series of temperature, soil moisture and FAPAR from 1999-2019 before (upper row) and after standardisation (lower row) for an exemplary pixel*

[Figure]

*Figure R2: Histograms of temperature, soil moisture and FAPAR from 1999-2019 before (upper row) and after standardisation (lower row) for an exemplary pixel*

Many time series are not normally distributed (according to the Shapiro-Wilk test the assumption of normality is rejected for 23.3%, 66.7%, 69,6% of the time series of deseasonalised temperature, soil moisture and FAPAR used in this study, based on a p-value of 0.05, respectively). However, normally distributed data is not a prerequesite for standardizing the time series to z-scores. This transformation has been applied in similar contexts for a variety of data sets (von Buttlar et al. 2018 and Orth et al. 2020, both published in *Biogeosciences*, Seddon et al. 2016, published in *Nature)* including non-normally distributed data such as precipitation, as it was carried out e.g. in the Ahlström et al. 2015, published in *Science*.

We repeated the entire analysis, standardizing by dividing by the interquartile range instead of the standard deviation. The differences in the results of the vulnerability analysis are negligible: statistical significance changes in 3 out of the 504 cases displayed for the land covers classes and subregions as despicted in Figs. 5, 6, A1 and A2 of the article. For all other combinations of land covers, subregions at the 12 months of the year, statistical significance remains unchanged. The standardisation using z-scores is statistical sound in the context of our study, but we can provide our findings based on division by the interquartile range instead of the standard deviation if required.

**b) Maturity of remotely sensed soil moisture data**

To address the uncertainties of the satellite-based soil moisture data set from ESA CCI, we additionally used the soil moisture reanalysis data set from ERA5 Land for the layers 0-7 cm, 7-28 cm and 28-100 cm (corresponding to the soil moisture data sets used in Nicolai-Shaw et al. 2017) to assess the soundness of our results. Both data sets show agreement in the overall patterns (see Appendix A). We only show the first layer (0-7 cm) from ERA5 Land in the submitted manuscript, because it is rather well correlated with deeper layers and there are only subtle changes in vulnerability between these layers. This is in line with findings by Orth et al. (2020) who also found only minor difference between surface and deep layers. In addition to Figure A1 and A2, we will add corresponding figures for the layers 7-28 cm and 28-100 cm, to enable the reader to assess how vulnerability to soil moisture varies within the top 100 cm of the soil according to the reanalysis data set from ERA5 Land. We appreciate that the additional usage of the soil moisture data set from ERA5 Land should be described more prominently in the manuscript (currently in lines 5-6, 80-81 and 200-202). We will further mention its usage in line 58 in the introduction.

The optimal hydrologic variable referring to ecosystem vulnerability is plant available water within the root zone. However, observations of this variable are not available for large spatial scales. Therefore, it is necessary to use either related observational data as proxies such as precipitation or satellite surface moisture data or data resulting from land surface models or assimilation of model and observations, all of which have specific constraints.

Soil moisture values are included in the land-data assimilation in the ERA5 Land reanalysis data set based on a point-wise Simplified Extended Kalman Filter (de Rosnay et al. 2013, Hersbach et al. 2020). Surface soil moisture is linked to root-zone soil moisture via soil water redistribution to deeper layers and therefore remotely sensed surface soil moisture is used for the approximation of root zone soil moisture in land data assimilation systems (Maggioni and Houser 2017, Orth et al. 2020). We are well aware that the dynamics of surface soil moisture are directly linked to wetting and drying processes, i.e. rainwater infiltration and soil evaporation. Thus, soil water dynamics of deeper soil layers show a dampened and delayed dynamics compared to the surface layer, but still there is a significant correlation between them (Akbar et al. 2018). ERA5 is the first ECMWF reanalysis data set, which includes satellite information for the derivation of soil moisture estimates (Hersbach et al. 2020).

We will add a further sentence on the delayed connection of surface and deeper soil layers in line 297:

"For example, it takes some time for the propagation of surface drying to deeper soil layers, because of the slow capillary flow (Berg et al. 2017)."

We further argue that using satellite-based soil moisture data in large-scale ecological applications is a state-of-the art method. We want to emphasize that remote sensing soil moisture products have significantly improved over the last years (Mohanty et al. 2017, Gruber et al. 2019) (see lines 199-200), are increasingly being used to assess the impact of droughts on plant productivity (Dorigo and de Jeu 2016) and the ESA CCI soil moisture has been applied succesfully in a large number of studies for ecological applications. For an overview we refer e.g. to Table 5 and section 4.3 in Dorigo et al. (2017). This shows that surface soil moisture – despite its undisputed limitation – can give valuable insights on the state of the ecosystems. Denissen et al. 2019 state that satellite surface soil moisture is well suited to infer the state of the vegetation and corresponding land-atmosphere interactions during climate extremes. The transition from an energy- to a water-limited regime is marked by the critical soil moisture regions (Denissen et al. 2019), which makes soil moisture an appropriate variable especially in Mediterranean environemnets (Szczypta et al. 2014). Chen et al. (2014) assessed variability in the NDVI over Australia and conclude that not only precipitation but also remotely sensed soil moisture data from ESA CCI can be a good predictor for vegetation growth. A further example where ESA CCI soil moisture is used for biospheric drought effect assessment can be found in Orth et al. (2020) (published in *Biogeosciences*). Nicolai-Shaw et al. (2017) emphasize that remotely sensed soil moisture is a valuable addition or might even be able to replace other soil moisture proxies for the investigation of land-vegetation-atmosphere-dynamics (see lines 54-55). The authors highlight the usage of remote sensing based soil moisture for the assessment of drought development. They found strong responses of grasslands to soil moisture droughts, while forests showed weaker responses and relate this to the shallower rooting depth of grasslands compared to forests.

We will add a sentence on this at line 298:

"Nicolai-Shaw et al. (2017) found that soil moisture data from ESA CCI was a good indicator for drought in grasslands, while forests exhibited weaker responses, probably due to access to deeper soil layers for forests compared to grasslands."

For a comparison of the ESA CCI soil moisture data set with gridded precipitation, see e.g. Dorigo et al. (2012, 2017). Furthermore, the ESA CCI soil moisture data set was even used to create a global precipitation product, which was compared to state-of-the-art precipitation data sets and showed relatively good performance (Ciabatta et al. 2018). We also would like to emphasize that ESA CCI soil moisture has been validated succesfully with in situ observations e.g. in Spain and France at depths from 5 to 30 cm (see lines 299-301; Albergel et al. 2013, Dorigo et al. 2015) and Turkey (Bulut et al. 2019).

Finally, precipitation does not directly translate into plant available water within the root zone (de Boeck et al. 2011; see lines 47-49) and is thus not necessarily superior to surface soil moisture data for the assessment of plant available soil water. Several other processes play a role, such as evapotranspiration (especially important in the Mediterranean), runoff, topography, soil properties and irrigation (Mohanty et al. 2017).

**c) Spatial aggregation of land cover data**

A general remark on the data used in this study: There is a certain mismatch between the desirable spatial scale for assessing ecological impacts and the actually available climatic and land cover data sets (Ummenhofer et al. 2017). So, while remote sensing offers consistent large spatial and temporal coverage, there remains a trade-off regarding their coarse spatial resolution.

A more detailed land cover classification map would be beneficial, but is not available for the Mediterreanean Basin to our knowledge (CORINE Land Cover would show greater detail but is only available for the European side of Mediterranean basin).

We argue that it is common to use such rather broad categories, which rather resemble plant functional types than specific plant communities, for the analysis of the impact of climatic extremes on ecosystems. Plant functional types are partially based on climatic preferences (Bonan 2016). While there are certain simplifications, their distinctions nevertheless provide valuable insights on key ecological properties. For example, when comparing broadleaf and needleleaf forests, the former have high photosynthetic rates and stomatal conductance, the latter have lower photosynthetic rates and conductance (Bonan 2016), and it has been shown, that they show differences in drought-induced growth reductions leading to tree mortality (Cailleret et al. 2016). As a further example, Teuling et al. (2010) investigate the differing responses to heatwaves for forests and herbaceous perennial vegetation using data from various European flux tower sites. Furthermore, dynamic global vegetation models are usually based on plant functional types and these models are built with the intent to investigate the feedback of ecosystems and climate (Bonan 2016).
Therefore, we argue that we use an adequate state of the art classification scheme for investigating interactions of ecosystems and climate at such spatial scales for our type of research.

Land cover classification schemes like the one in our study are commonly applied; examples of studies using the same or similar land cover classifications in such a context include Ceccherini et al. (2014), Baumbach et al. (2017) (published in *Biogeosciences*), Nicolai-Shaw et al. (2017) and Buitenwerf et al. (2018).

We would further like to point out that we only include grid points belonging to the Köppen-Geiger classes Csa ("Warm temperate climate with dry and hot summer") or Csb ("Warm temperate climate with dry and warm summer"), i.e. areas with alpine grasslands and semiarid steppes are not considered in the study.

**d) Ground truth**

We agree that validation with ground truth is generally desirable.

We would like to point out that all data sets have been validated using both ground truth and model data and underwent quality assessment see e.g. Dorigo et al. (2015), Dorigo et al. (2017) for ESA CCI, Sanchez-Zapero (2019), Fuster et al. (2020) for FAPAR and Hersbach et al. (2020) for ERA5. We would like to emphasize that ESA CCI soil moisture has been validated succesfully with ground observations e.g. in Spain, France (see lines 299-301; Albergel et al. 2013, Dorigo et al. 2015) and Turkey (Bulut et al. 2019) and the FAPAR has been validated with observation data from e.g. Tunisia, Italy, Spain and France primarily for a variety of crop types (Fuster et al. 2020). In addition to the already mentioned validation of ESA CCI soil moisture in Spain and France, we will mention its validation in Turkey and the validation of the FAPAR in line 301.

It is usually challenging to validate satellite data with ground truth, as ground truth does not exist in such a consistent form in space and time, as it would be desirable (Preimesberger et al. 2020). A full-fledged validation of our analysis remains therefore hardly feasible, as there is no ground truth data set available, which is consistently available for the representative land cover classes and subregions for all months of year for the entire time span to our knowledge. Because a comparison with in-situ data was not feasible in our case, we rather used ERA5 reanalysis data as an independent additional source for comparison to the soil moisture product from ESA CCI (see lines 199-202 and Appendix A), which is common practice in cases where sufficient in-situ data is not available (Preimesberger et al. 2020).

We argue that it is beyond the scope of our study to incorporate ground truth and it is common in the scientific literature to rely on validated data sets without carrying out an independent validation for the specific case study (for examples of similar studies without ground truth validation see e.g. van Oijen et al. 2013, Rolinski et al. 2015, Ivits et al. 2016, Baumbach et al. 2017, Nicolai-Shaw et al. 2017).

For the points raised regarding lines 231, 311 and 325 we refer to the corresponding sections below.

**e) Trend analysis**

We quantify vulnerability (after Rolinski et al. 2015) as the average deviation of the environmental variable under hazardous ecosystem conditions from values under non-hazardous ecosystem conditions for a specific time span (e.g. the vulnerability to temperature for all months of July in a grid point in central Spain regarding the time span 1999-2019). Thus, ecosystem vulnerability is based on the comparison of non-extreme to extreme conditions within a given time span, i.e. it is always related to a certain reference period and cannot be assigned for a single point in time. Therefore, a trend analysis investigating year-to-year changes is not directly feasible, but a trend anaylsis can be carried out by analysing several time spans, if time series are available for a sufficient length (e.g. comparing vulnerability for the periods 1999-2019, 1999-2024, 1999-2029).

A certain number of years encompassing a few extremes is required to obtain a meaningful baseline value. For only a small number of years, stochasticity is still too high for such a trend analysis. The time span we investigate has a length of 21 years from 1999 onwards (the year where the applied FAPAR product is first available), which is considered too short for a vulnerability trend analysis, as a reference period of 30 years is commonly suggested as a baseline in climatological settings (Stocker et al. 2013).

We will add the following text in line 282:

"The time series used here encompasses 21 years and is thus still too short for analyzing long-term trends. Nevertheless, our approach can potentially be used to monitor how vulnerability changes in future for the 12 months of the year by comparing vulnerability during different time spans if time series of sufficient length are available."

**References**

Ahlström, Anders; Raupach, Michael R.; Schurgers, Guy; Smith, Benjamin; Arneth, Almut; Jung, Martin et al. (2015): Carbon cycle. The dominant role of semi-arid ecosystems in the trend and variability of the land CO2 sink. *Science (New York, N.Y.)* 348 (6237), pp. 895–899. DOI: 10.1126/science.aaa1668.

Akbar, R., Short Gianotti, D., McColl, K.A., Haghighi, E., Salvucci, G.D., Entekhabi, D., 2018. Hydrological Storage Length Scales Represented by Remote Sensing Estimates of Soil Moisture and Precipitation. *Water Resour. Res.* 54, 1476–1492.

Albergel, C., Dorigo, W., Reichle, R.H., Balsamo, G., Rosnay, P. de, Muñoz-Sabater, J., Isaksen, L., Jeu, R. de, Wagner, W., 2013. Skill and Global Trend Analysis of Soil Moisture from Reanalyses and Microwave Remote Sensing. *J. Hydrometeor* 14, 1259–1277.

Baumbach, Lukas; Siegmund, Jonatan F.; Mittermeier, Magdalena; Donner, Reik V. (2017): Impacts of temperature extremes on European vegetation during the growing season. In *Biogeosciences* 14 (21), pp. 4891–4903. DOI: 10.5194/bg-14-4891-2017.

Berg, A., Sheffield, J., Milly, P.C.D., 2017. Divergent surface and total soil moisture projections under global warming. *Geophys Res Lett* 44, 236–244.

Boeck, H.J. de, Dreesen, F.E., Janssens, I.A., Nijs, I., 2011. Whole-system responses of experimental plant communities to climate extremes imposed in different seasons. *The New Phytologist* 189, 806–817.

Bonan, G.B., 2016. Ecological climatology: Concepts and applications, 3rd ed. Cambridge University Press, New York.

Buitenwerf, R., Sandel, B., Normand, S., Mimet, A., Svenning, J.-C., 2018. Land surface greening suggests vigorous woody regrowth throughout European semi-natural vegetation. *Global Change Biology* 24, 5789–5801.

Bulut, B., Yilmaz, M.T., Afshar, M.H., Şorman, A.Ü., Yücel, İ., Cosh, M.H., Şimşek, O., 2019. Evaluation of Remotely-Sensed and Model-Based Soil Moisture Products According to Different Soil Type, Vegetation Cover and Climate Regime Using Station-Based Observations over Turkey. *Remote Sensing* 11, 1875.

Cailleret, M., Jansen, S., Robert, E.M.R., Desoto, L., Aakala, T., Antos, J.A., Beikircher, B., Bigler, C., Bugmann, H., Caccianiga, M., Čada, V., Camarero, J.J., Cherubini, P., Cochard, H., Coyea, M.R., Čufar, K., Das, A.J., Davi, H., Delzon, S., Dorman, M., Gea-Izquierdo, G., Gillner, S., Haavik, L.J., Hartmann, H., Hereş, A.-M., Hultine, K.R., Janda, P., Kane, J.M., Kharuk, V.I., Kitzberger, T., Klein, T., Kramer, K., Lens, F., Levanic, T., Linares Calderon, J.C., Lloret, F., Lobo-Do-Vale, R., Lombardi, F., López Rodríguez, R., Mäkinen, H., Mayr, S., Mészáros, I., Metsaranta, J.M., Minunno, F., Oberhuber, W., Papadopoulos, A., Peltoniemi, M., Petritan, A.M., Rohner, B., Sangüesa-Barreda, G., Sarris, D., Smith, J.M., Stan, A.B., Sterck, F., Stojanović, D.B., Suarez, M.L., Svoboda, M., Tognetti, R., Torres-Ruiz, J.M., Trotsiuk, V., Villalba, R., Vodde, F., Westwood, A.R., Wyckoff, P.H., Zafirov, N., Martínez-Vilalta, J., 2017. A synthesis of radial growth patterns preceding tree mortality. *Global Change Biol* 23, 1675–1690.

Ceccherini, G., Gobron, N., Migliavacca, M., 2014. On the Response of European Vegetation Phenology to Hydroclimatic Anomalies. *Remote Sensing* 6, 3143–3169.

Chen, T., Jeu, R. de, Liu, Y.Y., van der Werf, G.R., Dolman, A.J., 2014. Using satellite based soil moisture to quantify the water driven variability in NDVI: A case study over mainland Australia. *Remote Sensing of Environment* 140, 330–338.

Ciabatta, L., Massari, C., Brocca, L., Gruber, A., Reimer, C., Hahn, S., Paulik, C., Dorigo, W., Kidd, R., Wagner, W., 2018. SM2RAIN-CCI: a new global long-term rainfall data set derived from ESA CCI soil moisture. *Earth Syst. Sci. Data* 10, 267–280.

De Rosnay, P., Drusch, M., Vasiljevic, D., Balsamo, G., Albergel, C., Isaksen, L., 2013. A simplified Extended Kalman Filter for the global operational soil moisture analysis at ECMWF. *Q.J.R. Meteorol. Soc.* 139, 1199–1213.

Denissen, J.M., Teuling, A.J., Reichstein, M., Orth, R., 2020. Critical Soil Moisture Derived From Satellite Observations Over Europe. *J. Geophys. Res. Atmos*. 125.

Dorigo, W.A., Gruber, A., Jeu, R. de, Wagner, W., Stacke, T., Loew, A., Albergel, C., Brocca, L., Chung, D., Parinussa, R.M., Kidd, R., 2015. Evaluation of the ESA CCI soil moisture product using ground-based observations. *Remote Sensing of Environment* 162, 380–395.

Dorigo, W., Jeu, R. de, 2016. Satellite soil moisture for advancing our understanding of earth system processes and climate change. *International Journal of Applied Earth Observation and Geoinformation* 48, 1–4.

Dorigo, W., Jeu, R. de, Chung, D., Parinussa, R., Liu, Y., Wagner, W., Fernández-Prieto, D., 2012. Evaluating global trends (1988-2010) in harmonized multi-satellite surface soil moisture. *Geophys Res Lett* 39.

Dorigo, W., Wagner, W., Albergel, C., Albrecht, F., Balsamo, G., Brocca, L., Chung, D., Ertl, M., Forkel, M., Gruber, A., Haas, E., Hamer, P.D., Hirschi, M., Ikonen, J., Jeu, R. de, Kidd, R., Lahoz, W., Liu, Y.Y., Miralles, D., Mistelbauer, T., Nicolai-Shaw, N., Parinussa, R., Pratola, C., Reimer, C., van der Schalie, R., Seneviratne, S.I., Smolander, T., Lecomte, P., 2017. ESA CCI Soil Moisture for improved Earth system understanding: State-of-the art and future directions. *Remote Sensing of Environment* 203, 185–215.

Fuster, B., Sánchez-Zapero, J., Camacho, F., García-Santos, V., Verger, A., Lacaze, R., Weiss, M., Baret, F., Smets, B., 2020. Quality Assessment of PROBA-V LAI, fAPAR and fCOVER Collection 300 m Products of Copernicus Global Land Service. *Remote Sensing* 12, 1017.

Gruber, Alexander; Scanlon, Tracy; van der Schalie, Robin; Wagner, Wolfgang; Dorigo, Wouter (2019): Evolution of the ESA CCI Soil Moisture climate data records and their underlying merging methodology. In *Earth Syst. Sci. Data* 11 (2), pp. 717–739. DOI: 10.5194/essd-11-717-2019.

Hersbach, H., Bell, B., Berrisford, P., Hirahara, S., Horányi, A., Muñoz-Sabater, J., Nicolas, J., Peubey, C., Radu, R., Schepers, D., Simmons, A., Soci, C., Abdalla, S., Abellan, X., Balsamo, G., Bechtold, P., Biavati, G., Bidlot, J., Bonavita, M., Chiara, G., Dahlgren, P., Dee, D., Diamantakis, M., Dragani, R., Flemming, J., Forbes, R., Fuentes, M., Geer, A., Haimberger, L., Healy, S., Hogan, R.J., Hólm, E., Janisková, M., Keeley, S., Laloyaux, P., Lopez, P., Lupu, C., Radnoti, G., Rosnay, P., Rozum, I., Vamborg, F., Villaume, S., Thépaut, J.-N., 2020. The ERA5 global reanalysis. *Q.J.R. Meteorol. Soc.* 146, 1999–2049.

Ivits, E., Horion, S., Erhard, M., Fensholt, R., 2016. Assessing European ecosystem stability to drought in the vegetation growing season. *Global Ecology and Biogeography* 25, 1131–1143.

Maggioni, V., Houser, P.R., 2017. Soil Moisture Data Assimilation. In: Park, S.K., Xu, L. (Eds.) Data Assimilation for Atmospheric, Oceanic and Hydrologic Applications (Vol. III). Springer International Publishing, Cham, pp. 195–217.

Mohanty, B.P., Cosh, M.H., Lakshmi, V., Montzka, C., 2017. Soil Moisture Remote Sensing: State-of-the-Science. *Vadose Zone Journal* 16, vzj2016.10.0105.

Nicolai-Shaw, N., Zscheischler, J., Hirschi, M., Gudmundsson, L., Seneviratne, S.I., 2017. A drought event composite analysis using satellite remote-sensing based soil moisture. *Remote Sensing of Environment* 203, 216–225.

Orth, R., Destouni, G., Jung, M., Reichstein, M., 2020. Large-scale biospheric drought response intensifies linearly with drought duration in arid regions. *Biogeosciences* 17, 2647–2656.

Phillips, Nathaniel D. (2018): YaRrr! The Pirate's Guide to R. Available online at https://bookdown.org/ndphillips/YaRrr/standardization-z-score.html.

Preimesberger, W., Scanlon, T., Su, C.-H., Gruber, A., Dorigo, W., 2021. Homogenization of Structural Breaks in the Global ESA CCI Soil Moisture Multisatellite Climate Data Record. *IEEE Trans. Geosci. Remote Sensing* 59, 2845–2862.

Rolinski, S., Rammig, A., Walz, A., Bloh, W. von, van Oijen, M., Thonicke, K., 2015. A probabilistic risk assessment for the vulnerability of the European carbon cycle to weather extremes: the ecosystem perspective. *Biogeosciences* 12, 1813–1831.

Sanchez-Zapero, J., 2019. Scientific quality evaluation: LAI, FAPAR, FCOVER Collection 1km V1 & V2.

Seddon, A.W.R., Macias-Fauria, M., Long, P.R., Benz, D., Willis, K.J., 2016. Sensitivity of global terrestrial ecosystems to climate variability. *Nature* 531, 229–232.

Stocker, T.F., Qin, D., Plattner, G.K., Tignor, M., Allen, S.K., Boschung, J., Nauels, A., Xia, Y., Bex, V., Midgley, P.M. (Eds.), 2013. Climate change 2013: The physical science basis Working Group I contribution to the Fifth assessment report of the Intergovernmental Panel on Climate Change. Cambridge University Press, New York, xi, 1535 pages.

Szczypta, C., Calvet, J.-C., Maignan, F., Dorigo, W., Baret, F., Ciais, P., 2014. Suitability of modelled and remotely sensed essential climate variables for monitoring Euro-Mediterranean droughts. *Geosci. Model Dev.* 7, 931–946.

Teuling, A.J., Seneviratne, S.I., Stöckli, R., Reichstein, M., Moors, E., Ciais, P., Luyssaert, S., van den Hurk, B., Ammann, C., Bernhofer, C., Dellwik, E., Gianelle, D., Gielen, B., Grünwald, T., Klumpp, K., Montagnani, L., Moureaux, C., Sottocornola, M., Wohlfahrt, G., 2010. Contrasting response of European forest and grassland energy exchange to heatwaves. *Nature Geosciences* 3, 722–727.

Ummenhofer, Caroline C.; Meehl, Gerald A. (2017): Extreme weather and climate events with ecological relevance: a review. *Philosophical transactions of the Royal Society of London. Series B, Biological sciences* 372 (1723). DOI: 10.1098/rstb.2016.0135.

van Oijen, M., Beer, C., Cramer, W., Rammig, A., Reichstein, M., Rolinski, S., Soussana, J.-F., 2013. A novel probabilistic risk analysis to determine the vulnerability of ecosystems to extreme climatic events. *Environ. Res. Lett.* 8, 15032.

von Buttlar, J., Zscheischler, J., Rammig, A., Sippel, S., Reichstein, M., Knohl, A., Jung, M., Menzer, O., Arain, M.A., Buchmann, N., Cescatti, A., Gianelle, D., Kieley, G., Law, B.E., Magliulo, V., Margolis, H., McCaughey, H., Merbold, L., Migliavacca, M., Montagnani, L., Oechel, W., Pavelka, M., Peichl, M., Rambal, S., Raschi, A., Scott, R.L., Vaccari, F.P., van Gorsel, E., Varlagin, A., Wohlfahrt, G., Mahecha, M.D., 2018. Impacts of droughts and extreme temperature events on gross primary production and ecosystem respiration: a systematic assessment across ecosystems and climate zones. *Biogeosciences* 15, 1293–1318.

---

## Author Response (AR2)

**Answer to Anonymous Referee #1**

We would like to thank the anonymous reviewer for the comments.

The reviewer raised concerns regarding a) the quality of soil moisture data, b) the spatial aggregation of land cover classes and Mediterranean subregions and c) the standardization technique.

The satellite-based soil moisture data set from ESA CCI is a state of the art data set, which underwent validation and quality assessment and has been used frequently for similar purposes in the scientific literature. Furthermore, the results from the reanalysis soil moisture data set from ERA5 Land are in agreement with the ones from ESA CCI in general. For details, we refer to our answer regarding the comments on lines 45-54 / 75-80.

The broad aggregation of land cover classes is common at such large spatial scales and a significantly more detailed land cover product is not available for the entire Mediterranean Basin to our knowledge. We argue that the classification of Mediterranean subregions allows visualizing certain large-scale patterns, while more detailed at pixel scale can be retrieved from the spatial maps provided in the article. For details, we refer to our answers regarding the comments on lines 69, Figure 5 and Figure 6.

The z-score standardization technique used here is commonly applied and does not bias our results. We additionally performed the analysis by dividing by the interquartile range instead of the standard deviation, however differences in the results of both approaches are negligible. For details, we refer to our answer regarding the comment on lines 100-105.

A detailed answer to all comments follows below. We refer to the lines corresponding to the submitted version.

**Major comments**

**Lines 45-54 / 75-80:**
To address the uncertainties of the satellite-based soil moisture data set from ESA CCI, we additionally used the soil moisture reanalysis data set from ERA5 Land for the layers 0-7 cm, 7-28 cm and 28-100 cm (corresponding to the soil moisture data sets used in Nicolai-Shaw et al. 2017) to assess the soundness of our results. Both data sets show agreement in the overall patterns (see Appendix A). We only show the first layer (0-7 cm) from ERA5 Land in the submitted manuscript, because it is rather well correlated with deeper layers and there are only subtle changes in vulnerability between these layers. This is in line with findings by Orth et al. (2020) who also found only minor difference between surface and deep layers. In addition to Figure A1 and A2, we will add corresponding figures for the layers 7-28 cm and 28-100 cm, to enable the reader to assess how vulnerability to soil moisture varies within the top 100 cm of the soil according to the reanalysis data set from ERA5 Land. We appreciate that the additional usage of the soil moisture data set from ERA5 Land should be described more prominently in the manuscript (currently in lines 5-6, 80-81 and 200-202). We will further mention its usage in line 58 in the introduction.

The optimal hydrologic variable referring to ecosystem vulnerability is plant available water within the root zone (see lines 294-295). However, observations of this variable are not available for large spatial scales. Therefore, it is necessary to use either related observational data as proxies such as precipitation or satellite surface moisture data or data resulting from land surface models or assimilation of model and observations, all of which have specific constraints.

Soil moisture values are included in the land-data assimilation in the ERA5 Land reanalysis data set based on a point-wise Simplified Extended Kalman Filter (de Rosnay et al. 2013, Hersbach et al. 2020). Surface soil moisture is linked to root-zone soil moisture via soil water redistribution to deeper layers and therefore remotely sensed surface soil moisture is used for the approximation of root zone soil moisture in land data assimilation systems (Maggioni and Houser 2017, Orth et al. 2020). We are well aware that the dynamics of surface soil moisture are directly linked to wetting and drying processes, i.e. rainwater infiltration and soil evaporation. Thus, soil water dynamics of deeper soil layers show a dampened and delayed dynamics compared to the surface layer, but still there is a significant correlation between them (Akbar et al. 2018). ERA5 is the first ECMWF reanalysis data set, which includes satellite information for the derivation of soil moisture estimates (Hersbach et al. 2020).

We will add a further sentence on the delayed connection of surface and deeper soil layers in line 297:

"For example, soil drying during summer affects primarily the top soil layer, while drying in deeper layers shows a lagged response, because upward capillary flow from these layers is comparatively slow (Berg et al., 2017)."

We further argue that using satellite-based soil moisture data in large-scale ecological applications is a state-of-the art method. We want to emphasize that remote sensing soil moisture products have significantly improved over the last years (Mohanty et al. 2017, Gruber et al. 2019) (see lines 199-200), are increasingly being used to assess the impact of droughts on plant productivity (Dorigo and de Jeu 2016) and the ESA CCI soil moisture has been applied succesfully in a large number of studies for ecological applications. For an overview we refer e.g. to Table 5 and section 4.3 in Dorigo et al. (2017). This shows that surface soil moisture – despite its undisputed limitation – can give valuable insights on the state of the ecosystems. Denissen et al. 2019 state that satellite surface soil moisture is well suited to infer the state of the vegetation and corresponding land-atmosphere interactions during climate extremes. The transition from an energy- to a water-limited regime is marked by the critical soil moisture regions (Denissen et al. 2019), which makes soil moisture an appropriate variable especially in Mediterranean environemnets (Szczypta et al. 2014). Chen et al. (2014) assessed variability in the NDVI over Australia and conclude that not only precipitation but also remotely sensed soil moisture data from ESA CCI can be a good predictor for vegetation growth. A further example where ESA CCI soil moisture is used for biospheric drought effect assessment can be found in Orth et al. (2020) (published in *Biogeosciences*). Nicolai-Shaw et al. (2017) emphasize that remotely sensed soil moisture is a valuable addition or might even be able to replace other soil moisture proxies for the investigation of land-vegetation-atmosphere-dynamics (see lines 54-55). The authors highlight the usage of remote sensing based soil moisture for the assessment of drought development. They found strong responses of grasslands to soil moisture droughts, while forests showed weaker responses and relate this to the shallower rooting depth of grasslands compared to forests.

We will add a sentence on this at line 298:

"Nicolai-Shaw et al. (2017) found that soil moisture data from ESA CCI was a good indicator for drought in grasslands, while forests exhibited weaker responses, probably due to access to deeper soil layers for forests compared to grasslands."

For a comparison of the ESA CCI soil moisture data set with gridded precipitation, see e.g. Dorigo et al. (2012, 2017). Furthermore, the ESA CCI soil moisture data set was even used to create a global precipitation product, which was compared to state-of-the-art precipitation data sets and showed relatively good performance (Ciabatta et al. 2018). We also would like to emphasize that ESA CCI soil moisture has been validated succesfully with in situ observations e.g. in Spain and France at depths

from 5 to 30 cm (see lines 299-301; Albergel et al. 2013, Dorigo et al. 2015) and Turkey (Bulut et al. 2019).

Finally, precipitation does not directly translate into plant available water within the root zone (de Boeck et al. 2011; see lines 47-49) and is thus not necessarily superior to surface soil moisture data for the assessment of plant available soil water. Several other processes play a role, such as evapotranspiration (especially important in the Mediterranean), runoff, topography, soil properties and irrigation (Mohanty et al. 2017).

**Line 69:**

A general remark on the data used in this study: There is a certain mismatch between the desirable spatial scale for assessing ecological impacts and the actually available climatic and land cover data sets (Ummenhofer et al. 2017). So, while remote sensing products offer consistent large spatial and temporal coverage, there remains a trade-off regarding their coarse spatial resolution.

A more detailed land cover classification map would be beneficial, but is not available for the Mediterreanean Basin to our knowledge (CORINE Land Cover would show greater detail but is only available for the European side of Mediterranean basin).

We argue that it is common to use such rather broad categories, which rather resemble plant functional types than specific plant communities, for the analysis of the impact of climatic extremes on ecosystems. Plant functional types are partially based on climatic preferences (Bonan 2016). While there are certain simplifications, their distinctions nevertheless provide valuable insights on key ecological properties. For example, when comparing broadleaf and needleleaf forests, the former have high photosynthetic rates and stomatal conductance, the latter have lower photosynthetic rates and conductance (Bonan 2016), and it has been shown, that they show differences in drought-induced growth reductions leading to tree mortality (Cailleret et al. 2016). As a further example, Teuling et al. 2010 investigate the differing responses to heatwaves for forests and herbaceous perennial vegetation using data from various European flux tower sites. Furthermore, dynamic global vegetation models are usually based on plant functional types and these models are built with the intent to investigate the feedback of ecosystems and climate (Bonan 2016).
Therefore, we argue that we use an adequate state of the art classification scheme for investigating interactions of ecosystems and climate at such spatial scales for our type of research.

Land cover classification schemes like the one in our study are commonly applied; examples of studies using the same or similar land cover classifications in such a context include Ceccherini et al. (2014), Baumbach et al. (2017) (published in *Biogeosciences*), Nicolai-Shaw et al. (2017) and Buitenwerf et al. (2018).

We would further like to point out that we only include grid points belonging to the Köppen-Geiger classes Csa ("Warm temperate climate with dry and hot summer") or Csb ("Warm temperate climate with dry and warm summer"), i.e. areas with alpine grasslands and semiarid steppes are not considered in the study.

**Lines 100-105:**

Calculating z-scores is a commonly applied standardisation technique (Phillips 2018). It does not lead to a bias in the tails of the distribution, but merely rescales the data. The purpose of this transformation here is to rescale soil moisture and temperature to a common scale, so they can be analysed and displayed jointly on this standardised scale (while maintaining their original distribution shape) (Orth

et al. 2020). For illustration, we show time series and histograms for an exemplary pixel for all three variables before and after the transformation below (Figs. R1 and R2).

[Figure]

Figure R1: Time series of temperature, soil moisture and FAPAR from 1999-2019 before (upper row) and after standardisation (lower row) for an exemplary pixel

[Figure]

Figure R2: Histograms of temperature, soil moisture and FAPAR from 1999-2019 before (upper row) and after standardisation (lower row) for an exemplary pixel

Many time series are not normally distributed (according to the Shapiro-Wilk test the assumption of normality is rejected for 23.3%, 66.7%, 69,6% of the time series of deseasonalised temperature, soil moisture and FAPAR used in this study, based on a p-value of 0.05, respectively). However, normally distributed data is not a prerequesite for standardizing the time series to z-scores. This transformation has been applied in similar contexts for a variety of data sets (von Buttlar et al. 2018 and Orth et al. 2020, both published in *Biogeosciences*, Seddon et al. 2016, published in *Nature*) including non-normally distributed data such as precipitation, as it was carried out e.g. in the Ahlström et al. 2015, published in *Science*.

We repeated the entire analysis, standardizing by dividing by the interquartile range instead of the standard deviation. The differences in the results of the vulnerability analysis are negligible: statistical

significance changes in 3 out of the 504 cases displayed for the land covers classes and subregions as despicted in Figs. 5, 6, A1 and A2 of the article. For all other combinations of land covers, subregions at the 12 months of the year, statistical significance remains unchanged. The standardisation using z-scores is statistical sound in the context of our study, but we can provide our findings based on division by the interquartile range instead of the standard deviation if required.

**Figure 5:**

High temperatures favor grain filling only for a certain temperature range. The relationship is nonlinear, so high temperature may favor crop growth until a certain threshold, where temperatures become too hot and limit crop growth (Hatfield and Prueger 2015). A variaty of crops is particularly vulnerable to temperature extremes during reproductive stages such as anthesis and grain filling (Luo 2011). The optimum and maximum temperature for grain filling for wheat is at 20.7 and 35.4°C, respectively (Porter and Gawith, 1999). We analysed the daily maximum temperature from ERA5 in May from 1999-2019 at all pixels with landcover "Crops (rainfed)" in the Mediterranean Basin west of 40°E and the optimum temperature is exceeded 69% of the time on average and the maximum temperature is exceeded at least once within this time span in 42% of the pixels. 31°C is stated as the physiological limit for wheat beyond which sterile grains are produced (Porter and Gawith, 1999). This temperature is exceeded on average 7% of the time and at least once within the time 1999-2019 for 83% of the grid points. This shows that relevant physiological temperature thresholds for crops are exceeded in the Mediterranean Basin in May and vulnerability to hot conditions is therefore plausible. For a detailed overview on crop sensitivity to temperature extremes during anthesis we refer to Hatfield and Prueger (2015).

We agree that crops such as maize and wheat are vulnerable to dry conditions in their reproductive phase (see e.g. Zhang and Oweis 1998, Daryanto et al., 2016). As we point out in lines 362-364, significant vulnerability to dry conditions in May is detected for various land cover classes in the ERA5 Land soil moisture data set, while there is no significant vulnerability in the ESA CCI data set. We will add a sentence in line 364, stating that the ERA5 Land soil moisture data set is presumably more realistic for the month of May.

"For land cover classes such as "Crops (rainfed)" vulnerability to dry conditions in May seems realistic, as various crops are prone to drought in their reproductive phase (Zhang and Oweis 1998, Daryanto et al., 2016), which indicates that ERA5 Land might give more plausible results for the month of May."

**Figure 6:**

In addition to the time series plots, spatial maps are provided to give further detail on the spatial patterns. Therefore, we argue that this division is still justified. It allow to quickly spot large-scale patterns (such as the prolonged vulnerability to hot and dry conditions in Turkey), while details which are not apparent in subregion aggregation can still be seen in the spatial maps.

**Minor comments**
**Line 20 and lines 26-30:**

Land abandonment has indeed led to increasing biomass and forest cover in the Mediterranean within the past decades (Spano et al. 2013, Peñuelas et al. 2017). We rather refer to increased tree mortality, growth reduction, extended fire risk, agricultural yield decline and vegetation shifts connected to increasing aridity and rising temperatures. Thank you for the remark, we will add more details on climatic impacts on ecosystems in the introduction.

We would further like to point out that ecosystems impacts are repeatedly addressed in the Discussion as well, e. g. in lines 232-237, 241-242, 261-274 and 312-320.

**Lines 24-25:**

Note that most of NE Spain is excluded from the study since it does not have a Mediterranean climate according to the Köppen-Geiger classification (see section 2.1 Study area and Fig. 1). Furthermore, we mention seasonal differences e.g. in lines 242-243.

**Lines 40-41:**

We will mention the linkage of extreme heat waves and Saharan air intrusions.

**Lines 45-46:**

We agree that soil moisture is important for crops during winter and spring and that it reaches its annual minimum in summer. High winter temperatures have been demonstrated to have both positive (see e.g. Sippel et al. 2018 for an example in Spain) and negative (see e.g. Ben-Ari et al. 2018 for an example in northern France) impacts on vegetation in combination with wet springs. This demonstrates that high winter temperatures can play a relevant role for crop productivity.

**Introduction in general:**

The research gap we address here is the inclusion of seasonality in the assessment of ecosystem vulnerability (see lines 31-33 and 59-63).

The introduction is structured in the following way: In the first two paragraphs (lines 20-30), we address characteristics of the Mediterranean Basin and the vulnerability of its ecosystems. In the third paragraph (lines 31-43), we introduce the importance of considering seasonality in the analysis of ecosystem vulnerability and discuss why it is important to investigate the impacts of climate anomalies on ecosystems in the Mediterranean. In the fourth paragraph (lines 44-58), we state that soil moisture is an important variable for ecosystem productivity and explain the potential of long-term satellite soil moisture products, which emerged within the last years for this purpose. We will add information on the FAPAR here in the resubmitted version. In the final paragraph (lines 59-63), we give details on our research aims.

**Lines 75-76:**

Long-term observation data sets such as the ESA CCI soil moisture data set usually contain inhomogeneities (Preimesberger et al. 2020). Single satellites only cover a limited time span; therefore inhomogeneities at transition times cannot be fully avoided. Such inconsistencies have been carefully investigated and the merging scheme of ESA CCI is considered to provide a viable long-term product (Su et al. 2016, Preimesberger et al. 2020).

**Lines 80-81:**

ESA CCI soil moisture is a satellite-based data set, while ERA5 Land is a reanalysis data set. We apply both data sets here for verification of our obtained results. Both data sets are commonly used in the

scientific literature and have been compared various times (see e.g. Preimesberger et al. 2020, Beck et al. 2021 or Albergel et al. 2013 and Dorigo et al. 2017 for their predecessor data sets).

**Lines 83-84:**

The bands of SPOT/VGT and PROBA-V cover similar spectral ranges (Smets et al. 2019). For the overlapping period from October 2013 to May 2014, the FAPAR from SPOT/VGT and PROBA-V show high agreement for all biome types (Verger et al. 2019).

**Line 86:**

We will write "Mediterranean climate" instead of "Mediterranean Basin" to be more precise. Note that we only investigate those regions of the Mediterranean Basin, which have a Mediterranean climate according to the Köppen-Geiger classification. We also mention that crops are affected by soil moisture in winter (see lines 241-242) in the discussion.

**Lines 85-96:**

We will move the information in this section to the introduction to improve readability, so that the background information is gathered jointly in this part of the manuscript.

**Error correction**

We noticed an error in our code. The percentiles defining hazardous conditions were wrongly indexed, leading to slightly different percentiles throughout. The changes are generally minor and all conclusions from our article can be inferred as before. Figures and text will be adjusted where needed. We would like to apologize for this inconvenience.

**References**

Ahlström, Anders; Raupach, Michael R.; Schurgers, Guy; Smith, Benjamin; Arneth, Almut; Jung, Martin et al. (2015): Carbon cycle. The dominant role of semi-arid ecosystems in the trend and variability of the land CO2 sink. **Science (New York, N.Y.)** 348 (6237), pp. 895–899. DOI: 10.1126/science.aaa1668.

Akbar, R., Short Gianotti, D., McColl, K.A., Haghighi, E., Salvucci, G.D., Entekhabi, D., 2018. Hydrological Storage Length Scales Represented by Remote Sensing Estimates of Soil Moisture and Precipitation. **Water Resour. Res.** 54, 1476–1492.

Albergel, C., Dorigo, W., Reichle, R.H., Balsamo, G., Rosnay, P. de, Muñoz-Sabater, J., Isaksen, L., Jeu, R. de, Wagner, W., 2013. Skill and Global Trend Analysis of Soil Moisture from Reanalyses and Microwave Remote Sensing. **J. Hydrometeor** 14, 1259–1277.

Baumbach, Lukas; Siegmund, Jonatan F.; Mittermeier, Magdalena; Donner, Reik V. (2017): Impacts of temperature extremes on European vegetation during the growing season. In **Biogeosciences** 14 (21), pp. 4891–4903. DOI: 10.5194/bg-14-4891-2017.

Beck, H.E., Pan, M., Miralles, D.G., Reichle, R.H., Dorigo, W.A., Hahn, S., Sheffield, J., Karthikeyan, L., Balsamo, G., Parinussa, R.M., van Dijk, A.I.J.M., Du, J., Kimball, J.S., Vergopolan, N., Wood, E.F., 2021. Evaluation of 18 satellite- and model-based soil moisture products using in situ measurements from 826 sensors. **Hydrology and Earth System Sciences** 25, 17–40.

Ben-Ari, T., Boé, J., Ciais, P., Lecerf, R., van der Velde, M., Makowski, D., 2018. Causes and implications of the unforeseen 2016 extreme yield loss in the breadbasket of France. ***Nature communications*** 9, 1627.

Berg, A., Sheffield, J., Milly, P.C.D., 2017. Divergent surface and total soil moisture projections under global warming. ***Geophys Res Lett*** 44, 236–244.

Boeck, H.J. de, Dreesen, F.E., Janssens, I.A., Nijs, I., 2011. Whole-system responses of experimental plant communities to climate extremes imposed in different seasons. ***The New Phytologist*** 189, 806–817.

Bonan, G.B., 2016. Ecological climatology: Concepts and applications, 3rd ed. Cambridge University Press, New York.

Buitenwerf, R., Sandel, B., Normand, S., Mimet, A., Svenning, J.-C., 2018. Land surface greening suggests vigorous woody regrowth throughout European semi-natural vegetation. ***Global Change Biology*** 24, 5789–5801.

Bulut, B., Yilmaz, M.T., Afshar, M.H., Şorman, A.Ü., Yücel, İ., Cosh, M.H., Şimşek, O., 2019. Evaluation of Remotely-Sensed and Model-Based Soil Moisture Products According to Different Soil Type, Vegetation Cover and Climate Regime Using Station-Based Observations over Turkey***. Remote Sensing*** 11, 1875.

Cailleret, M., Jansen, S., Robert, E.M.R., Desoto, L., Aakala, T., Antos, J.A., Beikircher, B., Bigler, C., Bugmann, H., Caccianiga, M., Čada, V., Camarero, J.J., Cherubini, P., Cochard, H., Coyea, M.R., Čufar, K., Das, A.J., Davi, H., Delzon, S., Dorman, M., Gea-Izquierdo, G., Gillner, S., Haavik, L.J., Hartmann, H., Hereş, A.-M., Hultine, K.R., Janda, P., Kane, J.M., Kharuk, V.I., Kitzberger, T., Klein, T., Kramer, K., Lens, F., Levanic, T., Linares Calderon, J.C., Lloret, F., Lobo-Do-Vale, R., Lombardi, F., López Rodríguez, R., Mäkinen, H., Mayr, S., Mészáros, I., Metsaranta, J.M., Minunno, F., Oberhuber, W., Papadopoulos, A., Peltoniemi, M., Petritan, A.M., Rohner, B., Sangüesa-Barreda, G., Sarris, D., Smith, J.M., Stan, A.B., Sterck, F., Stojanović, D.B., Suarez, M.L., Svoboda, M., Tognetti, R., Torres-Ruiz, J.M., Trotsiuk, V., Villalba, R., Vodde, F., Westwood, A.R., Wyckoff, P.H., Zafirov, N., Martínez-Vilalta, J., 2017. A synthesis of radial growth patterns preceding tree mortality. ***Global Change Biol*** 23, 1675–1690.

Ceccherini, G., Gobron, N., Migliavacca, M., 2014. On the Response of European Vegetation Phenology to Hydroclimatic Anomalies. ***Remote Sensing*** 6, 3143–3169.

Chen, T., Jeu, R. de, Liu, Y.Y., van der Werf, G.R., Dolman, A.J., 2014. Using satellite based soil moisture to quantify the water driven variability in NDVI: A case study over mainland Australia. ***Remote Sensing of Environment*** 140, 330–338.

Ciabatta, L., Massari, C., Brocca, L., Gruber, A., Reimer, C., Hahn, S., Paulik, C., Dorigo, W., Kidd, R., Wagner, W., 2018. SM2RAIN-CCI: a new global long-term rainfall data set derived from ESA CCI soil moisture. ***Earth Syst. Sci. Data*** 10, 267–280.

Daryanto, S., Wang, L., Jacinthe, P.-A., 2016. Global Synthesis of Drought Effects on Maize and Wheat Production. ***PloS one*** 11, e0156362.

De Rosnay, P., Drusch, M., Vasiljevic, D., Balsamo, G., Albergel, C., Isaksen, L., 2013. A simplified Extended Kalman Filter for the global operational soil moisture analysis at ECMWF. ***Q.J.R. Meteorol. Soc.*** 139, 1199–1213.

Denissen, J.M., Teuling, A.J., Reichstein, M., Orth, R., 2020. Critical Soil Moisture Derived From Satellite Observations Over Europe. ***J. Geophys. Res. Atmos***. 125.

Dorigo, W.A., Gruber, A., Jeu, R. de, Wagner, W., Stacke, T., Loew, A., Albergel, C., Brocca, L., Chung, D., Parinussa, R.M., Kidd, R., 2015. Evaluation of the ESA CCI soil moisture product using ground-based observations. ***Remote Sensing of Environment*** 162, 380–395.

Dorigo, W., Jeu, R. de, 2016. Satellite soil moisture for advancing our understanding of earth system processes and climate change. ***International Journal of Applied Earth Observation and Geoinformation*** 48, 1–4.

Dorigo, W., Jeu, R. de, Chung, D., Parinussa, R., Liu, Y., Wagner, W., Fernández-Prieto, D., 2012. Evaluating global trends (1988-2010) in harmonized multi-satellite surface soil moisture. ***Geophys Res Lett*** 39.

Dorigo, W., Wagner, W., Albergel, C., Albrecht, F., Balsamo, G., Brocca, L., Chung, D., Ertl, M., Forkel, M., Gruber, A., Haas, E., Hamer, P.D., Hirschi, M., Ikonen, J., Jeu, R. de, Kidd, R., Lahoz, W., Liu, Y.Y., Miralles, D., Mistelbauer, T., Nicolai-Shaw, N., Parinussa, R., Pratola, C., Reimer, C., van der Schalie, R., Seneviratne, S.I., Smolander, T., Lecomte,

P., 2017. ESA CCI Soil Moisture for improved Earth system understanding: State-of-the art and future directions. **Remote Sensing of Environment** 203, 185–215.

Gruber, Alexander; Scanlon, Tracy; van der Schalie, Robin; Wagner, Wolfgang; Dorigo, Wouter (2019): Evolution of the ESA CCI Soil Moisture climate data records and their underlying merging methodology. In **Earth Syst. Sci. Data** 11 (2), pp. 717–739. DOI: 10.5194/essd-11-717-2019.

Hatfield, J.L., Prueger, J.H., 2015. Temperature extremes: Effect on plant growth and development. **Weather and Climate Extremes** 10, 4–10.

Hersbach, H., Bell, B., Berrisford, P., Hirahara, S., Horányi, A., Muñoz-Sabater, J., Nicolas, J., Peubey, C., Radu, R., Schepers, D., Simmons, A., Soci, C., Abdalla, S., Abellan, X., Balsamo, G., Bechtold, P., Biavati, G., Bidlot, J., Bonavita, M., Chiara, G., Dahlgren, P., Dee, D., Diamantakis, M., Dragani, R., Flemming, J., Forbes, R., Fuentes, M., Geer, A., Haimberger, L., Healy, S., Hogan, R.J., Hólm, E., Janisková, M., Keeley, S., Laloyaux, P., Lopez, P., Lupu, C., Radnoti, G., Rosnay, P., Rozum, I., Vamborg, F., Villaume, S., Thépaut, J.-N., 2020. The ERA5 global reanalysis. **Q.J.R. Meteorol. Soc.** 146, 1999–2049.

Luo, Q., 2011. Temperature thresholds and crop production: a review. **Climatic change** 109, 583–598.

Maggioni, V., Houser, P.R., 2017. Soil Moisture Data Assimilation. In: Park, S.K., Xu, L. (Eds.) Data Assimilation for Atmospheric, Oceanic and Hydrologic Applications (Vol. III). Springer International Publishing, Cham, pp. 195–217.

Mohanty, B.P., Cosh, M.H., Lakshmi, V., Montzka, C., 2017. Soil Moisture Remote Sensing: State-of-the-Science. **Vadose Zone Journal** 16, vzj2016.10.0105.

Nicolai-Shaw, N., Zscheischler, J., Hirschi, M., Gudmundsson, L., Seneviratne, S.I., 2017. A drought event composite analysis using satellite remote-sensing based soil moisture. **Remote Sensing of Environment** 203, 216–225.

Orth, R., Destouni, G., Jung, M., Reichstein, M., 2020. Large-scale biospheric drought response intensifies linearly with drought duration in arid regions. **Biogeosciences** 17, 2647–2656.

Peñuelas, J., Sardans, J., Filella, I., Estiarte, M., Llusià, J., Ogaya, R., Carnicer, J., Bartrons, M., Rivas-Ubach, A., Grau, O., Peguero, G., Margalef, O., Pla-Rabés, S., Stefanescu, C., Asensio, D., Preece, C., Liu, L., Verger, A., Barbeta, A., Achotegui-Castells, A., Gargallo-Garriga, A., Sperlich, D., Farré-Armengol, G., Fernández-Martínez, M., Liu, D., Zhang, C., Urbina, I., Camino-Serrano, M., Vives-Ingla, M., Stocker, B., Balzarolo, M., Guerrieri, R., Peaucelle, M., Marañón-Jiménez, S., Bórnez-Mejías, K., Mu, Z., Descals, A., Castellanos, A., Terradas, J., 2017. Impacts of Global Change on Mediterranean Forests and Their Services. **Forests** 8, 463.

Phillips, Nathaniel D. (2018): YaRrr! The Pirate's Guide to R. Available online at https://bookdown.org/ndphillips/YaRrr/standardization-z-score.html.

Porter, J.R., Gawith, M., 1999. Temperatures and the growth and development of wheat: a review**. European Journal of Agronomy** 10, 23–36.

Preimesberger, W., Scanlon, T., Su, C.-H., Gruber, A., Dorigo, W., 2021. Homogenization of Structural Breaks in the Global ESA CCI Soil Moisture Multisatellite Climate Data Record. **IEEE Trans. Geosci. Remote Sensing** 59, 2845–2862.

Seddon, A.W.R., Macias-Fauria, M., Long, P.R., Benz, D., Willis, K.J., 2016. Sensitivity of global terrestrial ecosystems to climate variability. **Nature** 531, 229–232.

Sippel, S., El-Madany, T.S., Migliavacca, M., Mahecha, M.D., Carrara, A., Flach, M., Kaminski, T., Otto, F.E.L., Thonicke, K., Vossbeck, M., Reichstein, M., 2018. Warm Winter, Wet Spring, and an Extreme Response in Ecosystem Functioning on the Iberian Peninsula. **Bull. Amer. Meteor. Soc.** 99, S80-S85.

Smets, Bruno; Verger, A.; Camacho, F.; van der Goten, R.; Jacobs, T. (2019): Copernicus Global Land Operations "Vegetation and Energy". Product User Manual. Leaf Area Index (LAI) Fraction Of Absorbed Photosynthetically Active Radiation (FAPAR) Fraction Of Vegetation Cover (FCOVER) Collection 1km Version 2 Issue 1.33. *Copernicus Global Land Operations*. Available online at https://land.copernicus.eu/global/products/fapar, checked on 4/17/2020.

Spano, D., Snyder, R.L., Cesaraccio, C., 2013. Mediterranean Phenology. In: Schwartz, M.D. (Ed.) Phenology: An Integrative Environmental Science. Springer Netherlands, Dordrecht, pp. 173–196.

Su, C.-H., Ryu, D., Dorigo, W., Zwieback, S., Gruber, A., Albergel, C., Reichle, R.H., Wagner, W., 2016. Homogeneity of a global multisatellite soil moisture climate data record. *Geophys Res Lett* 43, 11,245-11,252.

Szczypta, C., Calvet, J.-C., Maignan, F., Dorigo, W., Baret, F., Ciais, P., 2014. Suitability of modelled and remotely sensed essential climate variables for monitoring Euro-Mediterranean droughts. *Geosci. Model Dev.* 7, 931–946.

Teuling, A.J., Seneviratne, S.I., Stöckli, R., Reichstein, M., Moors, E., Ciais, P., Luyssaert, S., van den Hurk, B., Ammann, C., Bernhofer, C., Dellwik, E., Gianelle, D., Gielen, B., Grünwald, T., Klumpp, K., Montagnani, L., Moureaux, C., Sottocornola, M., Wohlfahrt, G., 2010. Contrasting response of European forest and grassland energy exchange to heatwaves. *Nature Geosciences* 3, 722–727.

Ummenhofer, Caroline C.; Meehl, Gerald A. (2017): Extreme weather and climate events with ecological relevance: a review. In *Philosophical transactions of the Royal Society of London. Series B, Biological sciences* 372 (1723). DOI: 10.1098/rstb.2016.0135.

Verger, Aleixandre; Baret, Frederic; Weiss, Marie (2019): Copernicus Global Land Operations "Vegetation and Energy" "CGLOPS-1". Algorithm Theorethical Basis Document. Leaf Area Index (LAI) Fraction of Absorbed Photosynthetically Active Radiation (FAPAR) Fraction of green Vegetation Cover (FCover) Collection 1km Version 2 Issue I1.41. *Copernicus Global Land Operations*. Available online at https://land.copernicus.eu/global/products/fapar, checked on 4/17/2020.

von Buttlar, J., Zscheischler, J., Rammig, A., Sippel, S., Reichstein, M., Knohl, A., Jung, M., Menzer, O., Arain, M.A., Buchmann, N., Cescatti, A., Gianelle, D., Kiely, G., Law, B.E., Magliulo, V., Margolis, H., McCaughey, H., Merbold, L., Migliavacca, M., Montagnani, L., Oechel, W., Pavelka, M., Peichl, M., Rambal, S., Raschi, A., Scott, R.L., Vaccari, F.P., van Gorsel, E., Varlagin, A., Wohlfahrt, G., Mahecha, M.D., 2018. Impacts of droughts and extreme temperature events on gross primary production and ecosystem respiration: a systematic assessment across ecosystems and climate zones. *Biogeosciences* 15, 1293–1318.

Zhang, H., Oweis, T., 1999. Water–yield relations and optimal irrigation scheduling of wheat in the Mediterranean region. *Agricultural Water Management* 38, 195–211.

**Answer to Anonymous Referee #2**

We would like to thank the anonymous reviewer for the detailed feedback, which helped to improve the article further. We will address all raised points below. We refer to the lines corresponding to the submitted version.

**Major comments**

**Ground truth**

We agree that validation with ground truth is generally desirable.

We would like to point out that all data sets have been validated using both ground truth and model data and underwent quality assessment see e.g. Dorigo et al. (2015), Dorigo et al. (2017) for ESA CCI, Sanchez-Zapero (2019), Fuster et al. (2020) for FAPAR and Hersbach et al. (2020) for ERA5. We would like to emphasize that ESA CCI soil moisture has been validated succesfully with ground observations e.g. in Spain, France (see lines 299-301; Albergel et al. 2013, Dorigo et al. 2015) and Turkey (Bulut et al. 2019) and the FAPAR product has been validated with observation data from e.g. Tunisia, Italy, Spain and France primarily for a variety of crop types, as well as a deciduous broadleaf forest in Italy and a needle-leaf forest in Spain (Fuster et al. 2020). In addition to the already mentioned validation of ESA CCI soil moisture in Spain and France, we will mention its validation in Turkey and the validation of the FAPAR in line 301.

It is usually challenging to validate satellite data with ground truth, as ground truth does not exist in such a consistent form in space and time, as it would be desirable (Preimesberger et al. 2020). A full-fledged validation of our analysis remains therefore hardly feasible, as there is no ground truth data set available, which is consistently available for the representative land cover classes and subregions for all months of year for the entire time span to our knowledge. Because a comparison with in-situ data was not feasible in our case, we rather used ERA5 reanalysis data as an independent additional source for comparison to the soil moisture product from ESA CCI (see lines 199-202 and Appendix A), which is common practice in cases where sufficient in-situ data is not available (Preimesberger et al. 2020).

We argue that it is beyond the scope of our study to incorporate ground truth and it is common in the scientific literature to rely on validated data sets without carrying out an independent validation for the specific case study (for examples of similar studies without ground truth validation see e.g. van Oijen et al. 2013, Rolinski et al. 2015, Ivits et al. 2016, Baumbach et al. 2017, Nicolai-Shaw et al. 2017).

For the points raised regarding lines 231, 311 and 325 we refer to the corresponding sections below.

**Trend analysis**

We quantify vulnerability (after Rolinski et al. 2015) as the average deviation of the environmental variable under hazardous ecosystem conditions from values under non-hazardous ecosystem conditions for a specific time span (e.g. the vulnerability to temperature for all months of July in a grid point in central Spain regarding the time span 1999-2019). Thus, ecosystem vulnerability is based on the comparison of non-extreme to extreme conditions within a given time span, i.e. it is always related to a certain reference period and cannot be assigned for a single point in time. Therefore, a trend analysis investigating year-to-year changes is not directly feasible, but a trend analysis can be carried out by analysing several time spans, if time series are available for a sufficient length (e.g. comparing vulnerability for the periods 1999-2019, 1999-2024, 1999-2029).

A certain number of years encompassing a few extremes is required to obtain a meaningful baseline value. For only a small number of years, stochasticity is still too high for such a trend analysis. The time span we investigate has a length of 21 years from 1999 onwards (the year where the applied FAPAR product is first available), which is considered too short for a vulnerability trend analysis, as a reference period of 30 years is commonly suggested as a baseline in climatological settings (Stocker et al. 2013).

We will add the following text in line 282:

"The time series used here encompasses 21 years and is thus still too short for analyzing long-term trends. Nevertheless, our approach can potentially be used to monitor how vulnerability changes in future for all twelve months of the year by comparing vulnerability during different multi-year time spans if time series of sufficient length are available."

**Minor comments**

**Lines 90-96:**

To avoid redundancy between "Introduction" and "Methods" we will move the information on FAPAR and ESA CCI soil moisture in the second and third paragraph of 2.2 to the introduction at line 59 and remove redundant parts.

**Line 105:**

Yes, σ is calculated for the whole year. The months of the year have different variabilities and we aim to preserve this intraannual variability (whereas a monthly calculation of σ would artificially produce months with equal variability). We will add the word "year-round".

"dividing by the **year-round** standard deviation of the deseasonalised time series"

**Section 2.4:**

Van Oijen et al. (2013) (which our definition is based on) also denote that vulnerability is sometimes referred to as sensitivity. Weißhuhn et al. (2018) define sensitivity as a "measure of susceptibility" to a hazard and according to Ionescu et al. (2009) sensitivity is "characterising how much a system's state is affected by a change in its input". These definitions are applicable to the notion of vulnerability in our article. We investigate if extreme reductions in ecosystem productivity are linked to significant deviations in temperature and soil moisture. Hence, only if temperature or soil moisture deviations are related to low FAPAR values, significant ecosystem vulnerability will be detected.

Smith 2011 states, "we must be able to attribute the extreme ecological response to the period of climate extremity. [… This is] critical for elucidating what factors may contribute to differential sensitivity of ecosystems to climate extremes."
According to the framework by Smith 2011, vulnerability to extreme climatic events is defined as a climate extreme leading to an extreme ecological response. The definition used in our article involves extremeness in the response, as well as a significant deviation of the climatic driver (to the climatic driver during non-extreme ecosystem conditions). Therefore, our definition differs in that regard that it includes extremeness only for the ecological response, not necessarily for the driver (but extremes in the driver usually are significant deviations and thus they are included, so our definition is broader than the one by Smith 2011). In our case, ecosystem vulnerability rather shows if the ecosystem variable is susceptible to certain climatic conditions (which do not need to be extreme).

It should be noted that our approach is impact-based (see lines 132-133); following the definition of Rolinski et al. 2015 who "[...] define hazardous conditions from an ecosystem perspective to quantify the probability of weather conditions determining ecosystem vulnerability". This means, we are asking which are the climatic conditions leading to extreme ecosystem response (perspective from the ecosystem: define ecological extreme and attribute it to climatic drivers) rather than asking what are ecological impacts of climate extremes (perspective from the climatic driver: defining a climatological extreme and attribute it to the ecological response).

Risk is not assessed in our approach. This could be done in principal by using a qualitative instead of a distributional threshold (Rolinski et al. 2015). Risk is related to hazard probability, i.e. the proportion of exceedances of the hazard-threshold. In our case, this threshold is a percentile, which means each pixel has the same hazard probability (10%), so risk analysis is not directly applicable here. We chose to use a relative threshold (a percentile) rather than an absolute threshold, because it is not straightforward to determine a meaningful absolute threshold for extremeness of the FAPAR with validity for all land covers and subregions.
We contacted one of the authors of Rolinski et al. 2015, who confirmed that a percentile-based approach is an appropriate choice for our setting.

We will make the following changes to the manuscript:

We will replace lines 112-114

"The ecosystem vulnerability methodology serves to attribute drivers to their impact and identify whether a univariate or bivariate driver can be attributed to the respective impact."
by
"In the context of our study, ecosystem vulnerability depicts if ecosystems are susceptible or sensitive to a certain hazard. It allows to attribute states of low ecosystem productivity to certain climatic conditions by linking such states to corresponding deviations in temperature and soil moisture."

We will add in line 132:

"Every grid point has the same number of months with hazardous ecosystem conditions, i.e. the same risk of exceeding the threshold is assumed uniformly for all grid points."

We will move

"Our approach is impact-based, i.e. it focusses on the extremeness of the impact rather than the extremeness of the driver because this enables relating multiple drivers to a single outcome (Zscheischler et al., 2014, 2018)."

from line 132-133 to line 145 and add:

"According to the framework by Smith (2011), vulnerability to extreme climatic events is defined as a climate extreme leading to an extreme ecological response. Therefore, our definition differs in that regard that it comprises extremeness only for the ecological response, not necessarily for the climatic driver. The definition used here is broader than the one by Smith (2011), because it includes significant deviations of the driver variable in general, not only extremes. In our case, ecosystem vulnerability rather shows if the ecosystem variable is susceptible to certain climatic conditions (which do not need to be extreme). "

Line 171:
Yes, in each case 10% are classified as extreme. Extremes are defined by the FAPAR, not the driver variables (temperature and soil moisture). Therefore, these FAPAR extremes do not necessarily have

to be linked to any anomalies in the driver variables. So, in the case of sparse vegetation the subset of temperature at times with FAPAR extremes (times where FAPAR is below the 10% percentile) is not significantly different from the subset of temperature at other times without FAPAR extremes (times where FAPAR is above the 10% percentile). Our approach is designed this way – identifying first the impacts and then relate them to potential drivers (see lines 132-133) –, so it specifically leaves the option that the regarded driver variables are not relevant in certain cases (which is also an important finding).

You also additionally mention that vegetation might be dormant. This is another affect, which can occur. It is important to note that the 10% defined as extremes in each time series are not equally distributed within the months of the year. As pointed out above in the comment on line 171, σ is calculated for the entire time series, not seperately for each month. Therefore, months with higher variability are more likely to have a higher number of extremes. Because dormant months have low variability in the FAPAR, they thus will have few (if any) FAPAR extremes. Our approach is designed like this because this implicitly minimizes extremes outside of the growing season (without the need to explicitly define a growing season). See also lines 329-339 in the article.

We will write:
"Sparse vegetation is probably well adapted to high temperatures, as it never shows vulnerability to hot conditions, which means that temperature during extreme ecosystem conditions is not significantly higher than during non-extreme ecosystem conditions."

Figure 6:
Thank you for pointing this out. We will adjust it accordingly.

Line 222:
Thank you, yes we mean indeed "…or dry system".

Line 231:
In Fig. R1 we show three example plots for the Iberian Peninsula (first row), northwestern Africa (second row) and the southeastern Mediterranean (third row). The months of August are marked by red vertical lines.

We will delete the word "presumably" in line 231. We will add "and the FAPAR values are usually at their annual minimum at this time of the year." in line 232.

[Figure]

*Figure R3: Time series of FAPAR from 1999-2019 for three example plots for the Iberian Peninsula (first row), northwestern Africa (second row) and the southeastern Mediterranean (third row). The months of August are marked by red vertical lines.*

**Line 267:**

We will write:

"The sensitivity to heat varies with phenophase (Hatfield and Prueger, 2015) and the effect on the carbon cycle can differ seasonally. High temperatures might e.g. increase carbon uptake by advancing spring onset, but may lead to uptake reductions in summer (Piao et al., 2019)."

**Lines 311 / 325:**

We considered this and ran the analysis with various time lags and moving average lengths. The article in its current form includes already many components: a bivariate setting (temperature and soil moisture), soil moisture from two sources (satellite and reanalysis data), various land covers and subregions at all months of the year. Therefore, we investigate (3 driver variables x 12 months x (6 subregions + 8 land covers) 504 cases currently, which lead to several thousand cases if multiplied additionally with various time lags. This makes it challenging to add further complexity.

As we pointed out (lines 311-328), finding the optimal time lag is particularly cumbersome, as the optimal time lag might differ depending on the driver variable (temperature or soil moisture) and the specific land cover.

Therefore, we decided to use only one time lag in our article because a thorough analysis of the influence of time lags might add too much complexity to the article and makes it also challenging to display all these cases visually in a comprehensive way.

**Error correction**

We noticed an error in our code. The percentiles defining hazardous conditions were wrongly indexed, leading to slightly different percentiles throughout. The changes are generally minor and all conclusions from our article can be inferred as before. Figures and text will be adjusted where needed. We would like to apologize for this inconvenience.

**References**

Albergel, C., Dorigo, W., Reichle, R.H., Balsamo, G., Rosnay, P. de, Muñoz-Sabater, J., Isaksen, L., Jeu, R. de, Wagner, W., 2013. Skill and Global Trend Analysis of Soil Moisture from Reanalyses and Microwave Remote Sensing. *J. Hydrometeor* 14, 1259–1277.

Baumbach, L., Siegmund, J.F., Mittermeier, M., Donner, R.V., 2017. Impacts of temperature extremes on European vegetation during the growing season. *Biogeosciences* 14, 4891–4903.

Bulut, B., Yilmaz, M.T., Afshar, M.H., Şorman, A.Ü., Yücel, İ., Cosh, M.H., Şimşek, O., 2019. Evaluation of Remotely-Sensed and Model-Based Soil Moisture Products According to Different Soil Type, Vegetation Cover and Climate Regime Using Station-Based Observations over Turkey. *Remote Sensing* 11, 1875.

Dorigo, W.A., Gruber, A., Jeu, R. de, Wagner, W., Stacke, T., Loew, A., Albergel, C., Brocca, L., Chung, D., Parinussa, R.M., Kidd, R., 2015. Evaluation of the ESA CCI soil moisture product using ground-based observations. *Remote Sensing of Environment* 162, 380–395.

Dorigo, W., Wagner, W., Albergel, C., Albrecht, F., Balsamo, G., Brocca, L., Chung, D., Ertl, M., Forkel, M., Gruber, A., Haas, E., Hamer, P.D., Hirschi, M., Ikonen, J., Jeu, R. de, Kidd, R., Lahoz, W., Liu, Y.Y., Miralles, D., Mistelbauer, T., Nicolai-Shaw, N., Parinussa, R., Pratola, C., Reimer, C., van der Schalie, R., Seneviratne, S.I., Smolander, T., Lecomte, P., 2017. ESA CCI Soil Moisture for improved Earth system understanding: State-of-the art and future directions. *Remote Sensing of Environment* 203, 185–215.

Fuster, B., Sánchez-Zapero, J., Camacho, F., García-Santos, V., Verger, A., Lacaze, R., Weiss, M., Baret, F., Smets, B., 2020. Quality Assessment of PROBA-V LAI, fAPAR and fCOVER Collection 300 m Products of Copernicus Global Land Service. *Remote Sensing* 12, 1017.

Hersbach, H., Bell, B., Berrisford, P., Hirahara, S., Horányi, A., Muñoz-Sabater, J., Nicolas, J., Peubey, C., Radu, R., Schepers, D., Simmons, A., Soci, C., Abdalla, S., Abellan, X., Balsamo, G., Bechtold, P., Biavati, G., Bidlot, J., Bonavita, M., Chiara, G., Dahlgren, P., Dee, D., Diamantakis, M., Dragani, R., Flemming, J., Forbes, R., Fuentes, M., Geer, A., Haimberger, L., Healy, S., Hogan, R.J., Hólm, E., Janisková, M., Keeley, S., Laloyaux, P., Lopez, P., Lupu, C., Radnoti, G., Rosnay, P., Rozum, I., Vamborg, F., Villaume, S., Thépaut, J.-N., 2020. The ERA5 global reanalysis. *Q.J.R. Meteorol. Soc.* 146, 1999–2049.

Ionescu, C., Klein, R.J.T., Hinkel, J., Kavi Kumar, K.S., Klein, R., 2009. Towards a Formal Framework of Vulnerability to Climate Change. *Environ Model Assess* 14, 1–16.

Ivits, E., Horion, S., Erhard, M., Fensholt, R., 2016. Assessing European ecosystem stability to drought in the vegetation growing season. *Global Ecology and Biogeography* 25, 1131–1143.

Nicolai-Shaw, N., Zscheischler, J., Hirschi, M., Gudmundsson, L., Seneviratne, S.I., 2017. A drought event composite analysis using satellite remote-sensing based soil moisture. *Remote Sensing of Environment* 203, 216–225.

Preimesberger, W., Scanlon, T., Su, C.-H., Gruber, A., Dorigo, W., 2021. Homogenization of Structural Breaks in the Global ESA CCI Soil Moisture Multisatellite Climate Data Record. *IEEE Trans. Geosci. Remote Sensing* 59, 2845–2862.

Rolinski, S., Rammig, A., Walz, A., Bloh, W. von, van Oijen, M., Thonicke, K., 2015. A probabilistic risk assessment for the vulnerability of the European carbon cycle to weather extremes: the ecosystem perspective. *Biogeosciences* 12, 1813–1831.

Sanchez-Zapero, J., 2019. Scientific quality evaluation: LAI, FAPAR, FCOVER Collection 1km V1 & V2.

Smith, M.D., 2011. An ecological perspective on extreme climatic events: a synthetic definition and framework to guide future research. *Journal of Ecology* 99, 656–663.

Stocker, T.F., Qin, D., Plattner, G.K., Tignor, M., Allen, S.K., Boschung, J., Nauels, A., Xia, Y., Bex, V., Midgley, P.M. (Eds.), 2013. Climate change 2013: The physical science basis Working Group I contribution to the Fifth assessment report of the Intergovernmental Panel on Climate Change. Cambridge University Press, New York, xi, 1535 pages.

van Oijen, M., Beer, C., Cramer, W., Rammig, A., Reichstein, M., Rolinski, S., Soussana, J.-F., 2013. A novel probabilistic risk analysis to determine the vulnerability of ecosystems to extreme climatic events. *Environ. Res. Lett.* 8, 15032.

Weißhuhn, P., Müller, F., Wiggering, H., 2018. Ecosystem Vulnerability Review: Proposal of an Interdisciplinary Ecosystem Assessment Approach. *Environmental management* 61, 904–915.

---

## Author Response (AR3)

**Rebuttal letter**

We would like to thank the reviewers for their valuable feedback. Furthermore, we would also like to thank the editor and Co-Editor-in-Chief for providing a very thorough and detailed assessment of our manuscript.

The line references stated here are referring to the second revision of the manuscript unless stated otherwise.

**Reviewer 1**

**Land cover and subregion classification**

We are aware that compromises have to be made for aggregations at such large spatial scales. Such aggregations are common in the scientific literature, sometimes even comprising several climate zones (see e.g. van Oijen et al. 2014, Rolinski et al. 2015, Baumbach et al. 2017, all published in *Biogeosciences*). An aggregation at such a large spatial scale might not be sound for every single grid point it comprises, but it nevertheless can give insights which are valid for a large proportion of the included grid points and thus be representative.

We decided to give aggregations for subregions on the one hand side, and to complement this additionally with maps, so that spatial patterns within a certain subregion can still be identified by the reader.

The land cover classification used in our study allows distinguishing differences in ecosystem vulnerability between major plant functional types and can provide valuable insights on seasonal dissimilarities. We agree that a more detailed land cover classification scheme – which is not available for the Mediterranean Basin as a whole to our knowledge – might add further value in a future analysis.

We restricted our analysis to the climate categories Csa ("Warm temperate climate with dry and hot summer") and Csb ("Warm temperate climate with dry and warm summer"), in order to reduce heterogeneity in the seasonal cycle of vegetation activity in our study region. The reviewer was concerned that grid points with high vegetation activity in spring and grid points with high vegetation activity in summer might be mixed up, leading to a confounding signal. However, we would like to point out that high vegetation activity in summer is rare. Less than 1% of the grid points have their maximum in July or August and two thirds of these grid points are irrigated cropland.

We will address the compromises linked to large-scale aggregations of land covers and subregions and add the following paragraphs to the manuscript:

In lines 78-80:

The study area is constrained to all grid points in the Mediterranean Basin belonging to the Köppen-Geiger classes Csa ("Warm temperate climate with dry and hot summer") and Csb ("Warm temperate climate with dry and warm summer") (cf. Fig. 1) to ensure a certain level of comparability within the study area.

In lines 339-345:

The ESA CCI land cover product applied in this study is a state of the art data set; a more detailed data set is currently not available for the Mediterranean Basin as a whole. The ESA CCI land cover classification allows only for the differentiation of major plant functional types and future studies

might benefit from a more refined land cover classification scheme with a broader variety of land cover classes.

Furthermore, the subregions used in this study are not fully homogeneous and there is a certain variability within a given subregion. Thus, the patterns identified in this study (see Figs. 5 and 6) cannot always be inferred for an entire subregion. Therefore, the ecosystem vulnerability maps (Figs. 7 and B1) should be additionally examined for the identification of potentially deviating patterns within subregions.

**Differences in the ESA CCI and ERA5 Land soil moisture data sets**

The reviewer mentions deviations between the results obtained with the soil moisture data sets from ESA CCI and ERA5 Land. While the data sets often align well, they deviate for Northwestern Africa throughout the course of the year and for the Iberian Peninsula in the first half of the year. There is a section (Appendix A) in the article investigating similarities and differences of the data sets and this is discussed in lines 397-400 in the first revision of the manuscript (lines 409-412 in the second revision). This is an interesting finding in itself indicating that in certain regions at certain times of the year, the data sets are inconclusive. This information is crucial for a sound interpretation of the data and might help the data providers to address these inconsistencies.

**Data standardization**

Yes, there seems to be a misunderstanding regarding the standardization technique. The data are not fitted to a probability distribution as the reviewer assumes. For subtracting the mean and dividing by the standard deviation, it is not necessary to assume any underlying distribution. The SPI calculation mentioned by the reviewer is a two-step procedure where the data are fitted to a gamma distribution and then transformed to reflect a normal distribution. We carry out no such transformation. Z-score calculation simply rescales the data to a scale with mean 0 and standard deviation of 1 to be able to display various variables with different physical units. As stated before, we repeated the analysis using division by the interquartile range instead of division by standard deviation and the discrepancies in the results between both ways of calculation are negligible.

**Reviewer 2**
**Major suggestion**

We fully agree with the concern of the reviewer regarding statistical robustness. For this exact purpose, we also decided not to give statistical significances for single grid points (and restrained to showing only the magnitudes of their ecosystem vulnerability instead), but solely do so for aggregations of grid points. We agree that the value of a single grid point is not particularly robust because of the limited data available at this scale; therefore, the maps are primarily intended to provide the reader a more detailed visualisation of large-scale spatial patterns. Our major conclusions can all be inferred from the time series plots, relying on aggregations of grid points, rather than individual grid points. Also for land cover classes and subregions with a smaller number of grid points (< 100), there is often significant ecosystem vulnerability detected in the time series plots. Nevertheless, it is plausible that non-significant cases in some months of the year might be due to the relatively small amount of data available for these land cover classes and subregions.

We will address this in the manuscript text:

In lines 213-215:

Furthermore, certain land cover classes and subregions encompass a relatively small subset of grid points and thus non-significant ecosystem vulnerability might be due to data scarcity in some of these cases.

in line 227-230:

It is noteworthy that for a given grid point at a given month, only 21 observations are available. Therefore, the robustness of the magnitude of ecosystem vulnerability of individual grid points is limited and should thus be interpreted with care. The maps in Figs. 7 and B1 primarily aim to identify large-scale spatial patterns, but do not provide information on statistical significance at a grid point scale.

**Minor suggestion**

We updated the color scheme of Figs. 7 and B1 accordingly. They now match the color scheme shown in Figs. 3, 5, 6, A1, A2, A3, A4, A5 and A6.

**References**

Baumbach, L., Siegmund, J.F., Mittermeier, M., Donner, R.V., 2017. Impacts of temperature extremes on European vegetation during the growing season. *Biogeosciences* 14, 4891–4903.

Rolinski, S., Rammig, A., Walz, A., Bloh, W. von, van Oijen, M., Thonicke, K., 2015. A probabilistic risk assessment for the vulnerability of the European carbon cycle to weather extremes: the ecosystem perspective. *Biogeosciences* 12, 1813–1831.

van Oijen, M., Balkovi, J., Beer, C., Cameron, D.R., Ciais, P., Cramer, W., Kato, T., Kuhnert, M., Martin, R., Myneni, R., Rammig, A., Rolinski, S., Soussana, J.-F., Thonicke, K., van der Velde, M., Xu, L., 2014. Impact of droughts on the carbon cycle in European vegetation: a probabilistic risk analysis using six vegetation models. *Biogeosciences* 11, 6357–6375.